# Coordinated Proximal Policy Optimization

**Zifan Wu**
School of Computer Science and Engineering
Sun Yat-sen University, Guangzhou, China
`wuzf5@mail2.sysu.edu.cn`

**Chao Yu**[*]
School of Computer Science and Engineering
Sun Yat-sen University, Guangzhou, China
`yuchao3@mail.sysu.edu.cn`

**Deheng Ye**
Tencent AI Lab, Shenzhen, China
`dericye@tencent.com`

**Junge Zhang**[*]
Institute of Automation
Chinese Academy of Science, Beijing, China
`jgzhang@nlpr.ia.ac.cn`

**Haiyin Piao**
School of Electronic and Information
Northwestern Polytechnical University, Xian, China
`haiyinpiao@mail.nwpu.edu.cn`

**Hankz Hankui Zhuo**
School of Computer Science and Engineering
Sun Yat-sen University, Guangzhou, China
`zhuohank@mail.sysu.edu.cn`

## Abstract

We present Coordinated Proximal Policy Optimization (CoPPO), an algorithm that extends the original Proximal Policy Optimization (PPO) to the multi-agent setting. The key idea lies in the coordinated adaptation of step size during the policy update process among multiple agents. We prove the monotonicity of policy improvement when optimizing a theoretically-grounded joint objective, and derive a simplified optimization objective based on a set of approximations. We then interpret that such an objective in CoPPO can achieve dynamic credit assignment among agents, thereby alleviating the high variance issue during the concurrent update of agent policies. Finally, we demonstrate that CoPPO outperforms several strong baselines and is competitive with the latest multi-agent PPO method (i.e. MAPPO) under typical multi-agent settings, including cooperative matrix games and the StarCraft II micromanagement tasks.

## 1 Introduction

Cooperative Multi-Agent Reinforcement Learning (CoMARL) shows great promise for solving various real-world tasks, such as traffic light control (Wu et al., 2020), sensor network management (Sharma and Chauhan, 2020) and autonomous vehicle coordination (Yu et al., 2019). In such applications, a team of agents aim to maximize a joint expected utility through a single global reward. Since multiple agents coexist in a common environment and learn and adapt their behaviour concurrently, the arising non-stationary issue makes it difficult to design an efficient learning method (Hernandez-Leal et al., 2017; Papoudakis et al., 2019). Recently, a number of CoMARL

---

[*]Corresponding authors.

35th Conference on Neural Information Processing Systems (NeurIPS 2021).

methods based on Centralized Training Decentralized Execution (CTDE) (Foerster et al., 2016) have been proposed, including policy-based (Lowe et al., 2017; Foerster et al., 2018; Wang et al., 2020; Yu et al., 2021) and value-based methods (Sunehag et al., 2018; Rashid et al., 2018; Son et al., 2019; Mahajan et al., 2019). While generally having more stable convergence properties (Gupta et al., 2017; Song et al., 2019; Wang et al., 2020) and being naturally suitable for problems with stochastic policies (Deisenroth et al., 2013; Su et al., 2021), policy-based methods still receive less attention from the community and generally possess inferior performance against value-based methods, as evidenced in the StarCraft II benchmark (Samvelyan et al., 2019).

The performance discrepancy between policy-based and value-based methods can be largely attributed to the inadequate utilization of the centralized training procedure in the CTDE paradigm. Unlike value-based methods that directly optimize the policy via centralized training of value functions using extra global information, policy-based methods only utilize centralized value functions for state/action evaluation such that the policy can be updated to increase the likelihood of generating higher values (Sutton et al., 2000). In other words, there is an update lag between intermediate value functions and the final policy in policy-based methods, and merely coordinating over value functions is insufficient to guarantee satisfactory performance (Grondman et al., 2012; Fujimoto et al., 2018).

To this end, we propose the *Coordinated Proximal Policy Optimization* (CoPPO) algorithm, a multi-agent extension of PPO (Schulman et al., 2017), to directly coordinate over the agents' policies by dynamically adapting the step sizes during the agents' policy update processes. We first prove a relationship between a lower bound of joint policy performance and the update of policies. Based on this relationship, a monotonic joint policy improvement can be achieved through optimizing an ideal objective. To improve scalability and credit assignment, and to cope with the potential high variance due to non-stationarity, a series of transformations and approximations are then conducted to derive an implementable optimization objective in the final CoPPO algorithm. While originally aiming at monotonic joint policy improvement, CoPPO ultimately realizes a direct coordination over the policies at the level of each agent's policy update step size. Concretely, by taking other agents' policy update into consideration, CoPPO is able to achieve dynamic credit assignment that helps to indicate a proper update step size to each agent during the optimization procedure. In the empirical study, an extremely hard version of the *penalty game* (Claus and Boutilier, 1998) is used to verify the efficacy and interpretability of CoPPO. In addition, the evaluation on the StarCraft II micromanagement benchmark further demonstrates the superior performance of CoPPO against several strong baselines.

The paper is organized as follows: Section 2 provides a background introduction. Section 3 introduces the derivation process of CoPPO. Section 4 presents the experimental studies, and Section 5 reviews some related works. Finally, Section 6 concludes the paper.

## 2 Background

We model the fully cooperative MARL problem as a Dec-POMDP (Oliehoek and Amato, 2016) which is defined by a tuple $G = \langle N, S, \Omega, O, A, R, P, \gamma \rangle$. $N$ is the number of agents and $S$ is the set of true states of the environment. Agent $i \in \{1, \dots, N\}$ obtains its partial observation $o^i \in \Omega$ according to the observation function $O(s, i)$, where $s \in S$. Each agent has an action-observation history $\tau^i \in T \equiv (\Omega \times A)^*$. At each timestep, agent $i$ chooses an action $a^i \in A$ according to its policy $\pi(a^i|\tau^i)$, and we use $\boldsymbol{a}$ to denote the joint action $\{a^1, \dots, a^N\}$. The environment then returns the reward signal $R(s, \boldsymbol{a})$ that is shared by all agents, and shifts to the next state according to the transition function $P(s'|s, \boldsymbol{a})$. The joint action-value function induced by a joint policy $\boldsymbol{\pi}$ is defined as: $Q^{\boldsymbol{\pi}}(s_t, \boldsymbol{a}_t) = \mathbb{E}_{s_{t+1:\infty}, \boldsymbol{a}_{t+1:\infty}}[\sum_{t'=0}^{\infty} \gamma^{t'} R_{t+t'}|s_t, \boldsymbol{a}_t]$, where $\gamma \in [0, 1)$ is the discounted factor. We denote the joint action of agents other than agent $i$ as $\boldsymbol{a}^{-i}$, and $\boldsymbol{\pi}^{-i}$, $\boldsymbol{\tau}^{-i}$ follow a similar convention. Joint policy $\boldsymbol{\pi}$ can be parameterized by $\theta = \{\theta^1, \dots, \theta^N\}$, where $\theta^i$ is the parameter set of agent $i$'s policy. Our problem setting follows the CTDE paradigm (Foerster et al., 2016), in which each agent executes its policy conditioned only on the partially observable information, but the policies can be trained centrally by using extra global information.

**Value-based MARL** In CTDE value-based methods such as (Sunehag et al., 2018; Rashid et al., 2018; Son et al., 2019; Mahajan et al., 2019), an agent selects its action by performing an $\arg\max$ operation over the local action-value function, i.e. $\pi^i(\tau^i) = \arg\max_{a^i} Q^i(\tau^i, a^i)$. Without loss in

generality, the update rule for value-based methods can be formulated as follows:

$$\Delta\theta^i \propto \mathbb{E}_{\boldsymbol{\pi}}\left[\left(R(s,\boldsymbol{a}) + \max_{\boldsymbol{a}'} Q^{tot}(s,\boldsymbol{a}') - Q^{tot}(s,\boldsymbol{a})\right)\frac{\partial Q^{tot}}{\partial Q^i}\nabla_{\theta^i}Q^i(\tau^i,a^i)\right], \qquad (1)$$

where $\theta^i$ represents the parameter of $Q^i$, and $Q^{tot}$ is the global action-value function. The centralized training procedure enables value-based methods to factorize $Q^{tot}$ into some local action-values. Eq. (1) is essentially Q-learning if rewriting $\frac{\partial Q^{tot}}{\partial Q^i}\nabla_{\theta^i}Q^i(\tau^i,a^i)$ as $\nabla_{\theta^i}Q^{tot}$. The partial derivative is actually a credit assignment term that projects the step size of $Q^{tot}$ to that of $Q^i$ (Wang et al., 2020).

**Policy-based MARL** In CTDE policy-based methods such as (Foerster et al., 2018; Lowe et al., 2017; Wang et al., 2020; Yu et al., 2021), an agent selects its action from an explicit policy $\pi^i(a^i|\tau^i)$. The vanilla multi-agent policy gradient algorithm updates the policy using the following formula:

$$\Delta\theta^i \propto \mathbb{E}_{\boldsymbol{\pi}}\left[Q^{tot}_{\boldsymbol{\pi}}(s,\boldsymbol{a})\nabla_{\theta^i}\pi^i(a^i|\tau^i)\right]. \qquad (2)$$

In order to reduce the variance and address the credit assignment issue, COMA (Foerster et al., 2018) replace $Q^{tot}$ with the counterfactual advantage: $A^{\boldsymbol{\pi}}(s,\boldsymbol{a}) = Q^{\boldsymbol{\pi}}(s,(a^i,\boldsymbol{a}^{-i})) - \mathbb{E}_{\hat{a}^i\sim\pi^i}[Q^{\boldsymbol{\pi}}(s,(\hat{a}^i,\boldsymbol{a}^{-i}))]$. This implies that fixing the actions of other agents (i.e. $\boldsymbol{a}^{-i}$), an agent will evaluate the action it has actually taken (i.e. $a^i$) by comparing with the average effect of other actions it may have taken.

## 3 Coordinated Proximal Policy Optimization

In policy-based methods, properly limiting the policy update step size is proven to be effective in single-agent settings (Schulman et al., 2015a, 2017). In cases when there are multiple policies, it is also crucial for each agent to take other agents' update into account when adjusting its own step size. Driven by this insight, we propose the CoPPO algorithm to adaptively adjust the step sizes during the update of the policies of multiple agents.

### 3.1 Monotonic Joint Policy Improvement

The performance of joint policy $\boldsymbol{\pi}$ is defined as: $J(\boldsymbol{\pi}) \doteq \mathbb{E}_{\boldsymbol{a}\sim\boldsymbol{\pi},s\sim\rho^{\boldsymbol{\pi}}}\left[\sum_{t=0}^{\infty}\gamma^t R_{t+1}(s,\boldsymbol{a})\right]$, where $\rho^{\boldsymbol{\pi}}$ is the unnormalized discounted visitation frequencies when the joint actions are chosen from $\boldsymbol{\pi}$. Then the difference between the performance of two joint policies, say $\boldsymbol{\pi}$ and $\tilde{\boldsymbol{\pi}}$, can be expressed as the accumulation of the global advantage over timesteps (see Appendix A.1 for proof):

$$J(\tilde{\boldsymbol{\pi}}) - J(\boldsymbol{\pi}) = \mathbb{E}_{\boldsymbol{a}\sim\tilde{\boldsymbol{\pi}},s\sim\rho^{\tilde{\boldsymbol{\pi}}}}\left[A^{\boldsymbol{\pi}}(s,\boldsymbol{a})\right], \qquad (3)$$

where $A^{\boldsymbol{\pi}}(s,\boldsymbol{a}) = Q^{\boldsymbol{\pi}}(s,\boldsymbol{a}) - V^{\boldsymbol{\pi}}(s)$ is the joint advantage function. This equation indicates that if the joint policy $\boldsymbol{\pi}$ is updated to $\tilde{\boldsymbol{\pi}}$, then the performance will improve when the update increases the probability of taking "good" joint actions so that $\sum_{\boldsymbol{a}}\tilde{\boldsymbol{\pi}}(\boldsymbol{a}|s)A^{\boldsymbol{\pi}}(s,\boldsymbol{a}) > 0$ for every $s$. Modeling the dependency of $\rho^{\tilde{\boldsymbol{\pi}}}$ on $\tilde{\boldsymbol{\pi}}$ involves the complex dynamics of the environment, so we extend the approach proposed in (Kakade and Langford, 2002) to derive an approximation of $J(\tilde{\boldsymbol{\pi}})$, denoted as $\tilde{J}_{\boldsymbol{\pi}}(\tilde{\boldsymbol{\pi}})$:

$$\tilde{J}_{\boldsymbol{\pi}}(\tilde{\boldsymbol{\pi}}) \doteq J(\boldsymbol{\pi}) + \mathbb{E}_{\boldsymbol{a}\sim\tilde{\boldsymbol{\pi}},s\sim\rho^{\boldsymbol{\pi}}}\left[A^{\boldsymbol{\pi}}(s,\boldsymbol{a})\right]. \qquad (4)$$

Note that if the policy is differentiable, then $\tilde{J}_{\boldsymbol{\pi}}(\tilde{\boldsymbol{\pi}})$ matches $J(\tilde{\boldsymbol{\pi}})$ to first order (see Appendix A.2 for proof). Quantitatively, we measure the difference between two joint policies using the *maximum total variation divergence* (Schulman et al., 2015a), which is defined by: $D_{TV}^{\max}[\pi\|\tilde{\pi}] \doteq \max_s D_{TV}[\pi(\cdot|s)\|\tilde{\pi}(\cdot|s)]$, where $D_{TV}[\pi(\cdot|s)\|\tilde{\pi}(\cdot|s)] = \frac{1}{2}\int_{\mathcal{A}}|\pi(a|s) - \tilde{\pi}(a|s)|da$ (the definition for the discrete case is simply replacing the integral with a summation, and our results remain valid in such case). Using the above notations, we can derive the following theorem:

**Theorem 1.** *Let* $\epsilon = \max_{s,\boldsymbol{a}}|A^{\boldsymbol{\pi}}(s,\boldsymbol{a})|, \alpha_i = \sqrt{\frac{1}{2}D_{TV}^{\max}[\pi^i\|\tilde{\pi}^i]}, 1 \le i \le N$, *and* $N$ *be the total number of agents, then the error of the approximation in Eq. (4) can be explicitly bounded as follows:*

$$\left|J(\tilde{\boldsymbol{\pi}}) - \tilde{J}_{\boldsymbol{\pi}}(\tilde{\boldsymbol{\pi}})\right| \le 4\epsilon\left[\frac{1 - \gamma\prod_{i=1}^{N}(1 - \alpha_i)}{1 - \gamma} - 1\right]. \qquad (5)$$

*Proof.* See Appendix A.3. □

As shown above, the upper bound is influenced by $\alpha_i$, $N$ and $\epsilon$. By definition, we have $\alpha_i \leq 1$ and $\epsilon \geq 0$, thus the upper bound will increase when $\alpha_i$ increases for any $i$, implying that it becomes harder to make precise approximation when any individual of the agents dramatically updates their policies. Also, the growth in the number of agents can raise the difficulty for approximation. As for $\epsilon$, from Eq. (5) we can roughly conclude that a larger advantage value can cause higher approximation error, and this is reasonable because $\tilde{J}_{\boldsymbol{\pi}}(\tilde{\boldsymbol{\pi}})$ approximates $J(\tilde{\boldsymbol{\pi}})$ by approximating the expectation over $A^{\boldsymbol{\pi}}$. Transforming the inequality in Eq. (5) leads to $J(\tilde{\boldsymbol{\pi}}) \geq \tilde{J}_{\boldsymbol{\pi}}(\tilde{\boldsymbol{\pi}}) - 4\epsilon \left( \frac{1-\gamma \prod_{i=1}^{N}(1-\alpha_i)}{1-\gamma} - 1 \right)$. Thus the joint policy can be iteratively updated by:

$$\boldsymbol{\pi}_{new} = \arg\max_{\tilde{\boldsymbol{\pi}}} \left[ \tilde{J}_{\boldsymbol{\pi}_{old}}(\tilde{\boldsymbol{\pi}}) - 4\epsilon \left( \frac{1-\gamma \prod_{i=1}^{N}(1-\alpha_i)}{1-\gamma} - 1 \right) \right]. \tag{6}$$

Eq. (6) involves a complete search in the joint observation space and action space for computing $\epsilon$ and $\alpha_i$, making it difficult to be applied to large-scale settings. In the next subsection, several transformations and approximations are employed to this objective to achieve better scalability.

### 3.2 The Final Algorithm

Notice that the complexity of optimizing the objective in Eq. (6) mainly lies in the second term, i.e. $4\epsilon \left( \frac{1-\gamma \prod_{i=1}^{N}(1-\alpha_i)}{1-\gamma} - 1 \right)$. While $\epsilon$ has nothing to do with $\tilde{\boldsymbol{\pi}}$, it suffices to control this second term only by limiting the variation divergence of agents' policies (i.e. $\alpha_i$), because it increases monotonically as $\alpha_i$ increases. Then the objective is transformed into $\tilde{J}_{\boldsymbol{\pi}_{old}}(\tilde{\boldsymbol{\pi}})$ that can be optimized subject to a trust region constraint: $\alpha_i \leq \delta, i = 1, \ldots, N$. For higher scalability, $\alpha_i$ can be replaced by the mean Kullback-Leibler Divergence between agent $i$'s two consecutive policies, i.e. $\mathbb{E}_{s \sim \rho^{\boldsymbol{\pi}}} \left[ D_{KL}[\pi^i(\cdot|\tau^i)||\tilde{\pi}^i(\cdot|\tau^i)] \right]$.

As proposed in (Schulman et al., 2015a), solving the above trust region optimization problem requires repeated computation of Fisher-vector products for each update, which is computationally expensive in large-scale problems, especially when there are multiple constraints. In order to reduce the computational complexity and simplify the implementation, importance sampling can be used to incorporate the trust region constraints into the objective of $\tilde{J}_{\boldsymbol{\pi}_{old}}(\tilde{\boldsymbol{\pi}})$, resulting in the maximization of $\mathbb{E}_{\boldsymbol{a} \sim \boldsymbol{\pi}_{old}} \left[ \min \left( \boldsymbol{r} A^{\boldsymbol{\pi}_{old}}, \text{clip} \left( \boldsymbol{r} A^{\boldsymbol{\pi}_{old}}, 1-\epsilon, 1+\epsilon \right) \right) \right]$ w.r.t $\boldsymbol{\pi}$, where $\boldsymbol{r} = \frac{\boldsymbol{\pi}(\boldsymbol{a}|s)}{\boldsymbol{\pi}_{old}(\boldsymbol{a}|s)}$, and $A^{\boldsymbol{\pi}_{old}}(s, \boldsymbol{a})$ is denoted as $A^{\boldsymbol{\pi}_{old}}$ for brevity. The clip function prevents the joint probability ratio from going beyond $[1-\epsilon, 1+\epsilon]$, thus approximately limiting the variation divergence of the joint policy. Since the policies are independent during the fully decentralized execution, it is reasonable to assume that $\boldsymbol{\pi}(\boldsymbol{a}|\boldsymbol{\tau}) = \prod_{i=1}^{N} \pi^i(a^i|\tau^i)$. Based on this factorization, the following objective can be derived:

$$\underset{\theta^1, \ldots, \theta^N}{\text{maximize}} \ \mathbb{E}_{\boldsymbol{a} \sim \boldsymbol{\pi}_{old}} \left\{ \min \left[ \left( \prod_{j=1}^{N} r^j \right) A^{\boldsymbol{\pi}}, \text{clip} \left( \left( \prod_{j=1}^{N} r^j \right), 1-\epsilon, 1+\epsilon \right) A^{\boldsymbol{\pi}} \right] \right\}, \tag{7}$$

where $\theta^j$ is the parameter of agent $j$'s policy, and $r^j = \frac{\pi^j(a^j|\tau^j;\theta^j)}{\pi_{old}^j(a^j|\tau^j;\theta_{old}^j)}$. While $A^{\boldsymbol{\pi}}$ is defined as $Q^{\boldsymbol{\pi}}(s, \boldsymbol{a}) - V^{\boldsymbol{\pi}}(s)$, the respective contribution of each individual agent cannot be well distinguished. To enable credit assignment, the joint advantage function is decomposed to some local ones of the agents as: $A^{\boldsymbol{\pi}}(s, \boldsymbol{a}) = \sum_{i=1}^{N} c^i \cdot A^i(s, (a^i, \boldsymbol{a}^{-i}))$, where $A^i(s, (a^i, \boldsymbol{a}^{-i})) = Q^{\boldsymbol{\pi}}(s, (a^i, \boldsymbol{a}^{-i})) - \mathbb{E}_{\hat{a}^i}[Q^{\boldsymbol{\pi}}(s, (\hat{a}^i, \boldsymbol{a}^{-i}))]$ is the counterfactual advantage and $c^i$ is a non-negative weight.

During each update, multiple epochs of optimization are performed on this joint objective to improve sample efficiency. Due to the non-negative decomposition of $A^{\boldsymbol{\pi}}$, there is a monotonic relationship between the global optimum and the local optima, suggesting a transformation from the joint objective to local objectives (see Appendix A.4 for the proof). The optimization of Eq. (7) then can be transformed to maximizing each agent's own objective:

$$L^i(\boldsymbol{\theta}) = \mathbb{E}_{\boldsymbol{a} \sim \boldsymbol{\pi}_{old}} \left\{ \min \left[ \left( \prod_{j \neq i} r^j \right) r^i A^i, \text{clip} \left( \left( \prod_{j \neq i} r^j \right) r^i, 1-\epsilon, 1+\epsilon \right) A^i \right] \right\}. \tag{8}$$

However, the ratio product in Eq. (8) raises a potential risk of high variance due to Proposition 1:

**Proposition 1.** *Assuming that the agents are fully independent during execution, then the following inequality holds:*

$$\text{Var}_{\boldsymbol{a}^{-i} \sim \boldsymbol{\pi}_{old}^{-i}} \left[ \prod_{j \neq i} r^j \right] \geq \prod_{j \neq i} \text{Var}_{a^j \sim \pi_{old}^j} \left[ r^j \right]. \tag{9}$$

*Proof.* See Appendix A.5. □

According to the inequality above, the variance of the product grows at least exponentially with the number of agents. Intuitively, the existence of other agents' policies introduces instability in the estimate of each agent's policy gradient. This may further cause suboptimatlity in individual policies due to the centralized-decentralized mismatch issue mentioned in (Wang et al., 2020). To be concrete, when $A^i > 0$, the external $\min$ operation in Eq. (8) can prevent the gradient of $L^i(\boldsymbol{\theta})$ from exploding when $\prod_{j \neq i} r^j$ is large, thus limiting the variance raised from other agents; but when $A^i < 0$, the gradient can grow rapidly in the negative direction, because $L^i(\boldsymbol{\theta}) = \mathbb{E}\left[ \left( \prod_{j \neq i} r^j \right) r^i A^i \right]$ when $\left( \prod_{j \neq i} r^j \right) r^i \geq 1 + \epsilon$. Moreover, the learning procedure in Eq. (8) can cause a potential exploration issue, that is, different agents might be granted unequal opportunities to update their policies. Consider a scenario when the policies of some agents except agent $i$ are updated rapidly, and thus the product of these agents' ratios might already be close to the clipping threshold, then a small optimization step of agent $i$ will cause $\prod_{j=1}^{N} r^j$ to reach the threshold and thus being clipped. In this case, agent $i$ has no chance to update its policy, while other agents have updated their policies significantly, leading to unbalanced exploration among the agents. To address the above issues, we propose a double clipping trick to modify Eq. (8) as follows:

$$L^i(\boldsymbol{\theta}) = \mathbb{E}_{\boldsymbol{a} \sim \boldsymbol{\pi}_{old}} \left\{ \min \left[ g(\boldsymbol{r}^{-i}) r^i A^i, \text{clip} \left( g(\boldsymbol{r}^{-i}) r^i, 1 - \epsilon_1, 1 + \epsilon_1 \right) A^i \right] \right\}, \tag{10}$$

where $g(\boldsymbol{r}^{-i}) = \text{clip}(\prod_{j \neq i} r^j, 1 - \epsilon_2, 1 + \epsilon_2)$, $\epsilon_2 < \epsilon_1$. In Eq. (8), the existence of $\prod_{j \neq i} r^j$ imposes an influence on the objective of agent $i$ through a weight of $\prod_{j \neq i} r^j$. Therefore, the clipping on $\prod_{j \neq i} r^j$ ensures that the influence from the update of other agents on agent $i$ is limited to $[1 - \epsilon_2, 1 + \epsilon_2]$, thus controlling the variance caused by other agents. Note that from the theoretical perspective, clipping separately on each individual probability ratio (i.e. $\prod_{j=1}^{N} \text{clip}(r^j, \cdot, \cdot)$) can also reduce the variance. Nevertheless, the empirical results show that clipping separately performs worse than clipping jointly. The detailed results and analysis for this comparison are presented in Appendix D.2.1. In addition, this trick also prevents the update step of each agent from being too small, because $r^i$ in Eq. (10) can at least increase to $\frac{1 + \epsilon_1}{1 + \epsilon_2} r^i$ or decrease to $\frac{1 - \epsilon_1}{1 - \epsilon_2} r^i$ before being clipped in each update.

In the next subsection, we will show that the presence of other agents' probability ratio also enables a dynamic credit assignment among the agents in order to promote coordination, and thus the inner clipping threshold (i.e. $\epsilon_2$) can actually function as a balance factor to trade off between facilitating coordination and reducing variance, which will be studied empirically in Section 4. A similar trick was proposed in (Ye et al., 2020a,b) to handle the variance induced by distributed training in the single-agent setting. Nonetheless, since multiple policies are updated in different directions in MARL, the inner clipping here is carried out on the ratio product of other agents instead of the entire ratio product, in order to distinguish the update of different agents. The overall CoPPO algorithm with the double clipping trick is shown in Appendix B.

### 3.3   Interpretation: Dynamic Credit Assignment

The COMA (Foerster et al., 2018) algorithm tries to address the credit assignment issue in CoMARL using the counterfactual advantage. Nevertheless, miscoordination and suboptimatlity can still arise since the credit assignment in COMA is conditioned on the fixed actions of other agents, but these actions are continuously changing and thus cannot precisely represent the actual policies. While CoPPO also makes use of the counterfactual advantage, the overall update of other agents is taken into account dynamically during the multiple epochs in each update. This process can adjust the

advantage value in a coordinated manner and alleviate the miscoordination issue caused by the fixation of other agents' joint action. Note that the theoretical reasoning for CoPPO in Section 3.1 and 3.2 originally aims at monotonic joint policy improvement, yet the resulted objective ultimately realizes coordination among agents through a dynamic credit assignment among the agents in terms of coordinating over the step sizes of the agents' policies.

To illustrate the efficacy of this dynamic credit assignment, we make an analysis on the difference between CoPPO and MAPPO (Yu et al., 2021) which generalizes PPO to multi-agent settings simply by centralizing the value functions with an optimization objective of $\mathbb{E}_{\boldsymbol{\pi}_{old}}\left[\min\left[r_k^i A^i, \text{clip}\left(r_k^i, 1-\epsilon, 1+\epsilon\right)\right]\right]$, which is a lower bound of $\mathbb{E}_{\boldsymbol{\pi}_{old}}\left[r_k^i A^i\right]$ where $r_k^i$ represents the probability ratio of agent $i$ at the $k_{th}$ optimization epoch during each update. Denoting $\left(\prod_{j\neq i} r_k^j\right) A^i$ as $\tilde{A}_k^i$, the CoPPO objective then becomes approximately a lower bound of $\mathbb{E}_{\boldsymbol{\pi}_{old}}\left[r_k^i \tilde{A}_k^i\right]$. The discussion then can be simplified to analyzing the two lower bounds (see Appendix A.6 for the details of this simplification).

Depending on whether $A^i > 0$ and whether $\prod_{j\neq i} r_k^j > 1$, four different cases can be classified. For brevity, only two of them are discussed below while the rest are similar. The initial parameters of the two methods are assumed to be the same.

Case (1): $A^i > 0$, $\prod_{j\neq i} r_k^j > 1$. In this case $\left|\tilde{A}_k^i\right| > \left|A^i\right|$, thus $\left\|\tilde{A}_k^i \nabla r_k^i\right\| > \left\|A^i \nabla r_k^i\right\|$, indicating that CoPPO takes a larger update step towards increasing $\pi^i(a^i|\tau^i)$ than MAPPO does. Concretely, $A^i > 0$ means that $a^i$ is considered (by agent $i$) positive for the whole team when fixing $\boldsymbol{a}^{-i}$ and under similar observations. Meanwhile, $\prod_{j\neq i} r_k^j > 1$ implies that after this update epoch, $\boldsymbol{a}^{-i}$ are overall more likely to be performed by the other agents when encountering similar observations (see Appendix A.7 for the details). This makes fixing $\boldsymbol{a}^{-i}$ more reasonable when estimating the advantage of $a^i$, thus explaining CoPPO's confidence to take a larger update step.

Case (2): $A^i < 0$, $\prod_{j\neq i} r_k^j < 1$. Similarly, in this case $\left|\tilde{A}_k^i\right| < \left|A^i\right|$ and hence $\left\|\tilde{A}_k^i \nabla r_k^i\right\| < \left\|A^i \nabla r_k^i\right\|$, indicating that CoPPO takes a smaller update step to decrease $\pi^i(a^i|\tau^i)$ than MAPPO does. To be specific, $a^i$ is considered (by agent $i$) to have a negative effect on the whole team since $A^i < 0$, and $\prod_{j\neq i} r_k^j < 1$ suggests that after this optimization epoch, other agents are overall less likely to perform $\boldsymbol{a}^{-i}$ given similar observations. While the evaluation of $a^i$ is conditioned on $\boldsymbol{a}^{-i}$, it is reasonable for agent $i$ to rethink the effect of $a^i$ and slow down the update of decreasing the probability of taking $a^i$, thus giving more chance for this action to be evaluated.

It is worth noting that $\tilde{A}_k^i$ continues changing throughout the K epochs of update and yields dynamic adjustments in the step size, while $A^i$ will remain the same during each update. Therefore, $\tilde{A}_k^i$ can be interpreted as a dynamic modification of $A^i$ by taking other agents' update into consideration.

# 4 Experiments

In this section, we evaluate CoPPO on a modified matrix penalty game and the StarCraft Multi-Agent Challenge (SMAC) (Samvelyan et al., 2019). The matrix game results enable interpretative observations, while the evaluations on SMAC verify the efficacy of CoPPO in more complex domains.

## 4.1 Cooperative Matrix Penalty Game

The *penalty game* is a representative of problems with miscoordination penalties and multiple equilibria selection among optimal joint actions. It has been used as a challenging test bed for evaluating CoMARL algorithms (Claus and Boutilier, 1998; Spiros and Daniel, 2002). To further increase the difficulty of achieving coordination, we modify the two-player *penalty game* to four agents with nine actions for each agent. The agents will receive a team reward of 50 when they have played the same action, but be punished by -50 if any three agents have acted the same while the other does not. In all other cases, the reward is -40 for all the agents. The *penalty game* provides a verifying metaphor to show the importance of adaptive adjustment in the agent policies in order to achieve efficient coordinated behaviors. Thinking of the case when the agents have almost reached one of the optimal joint actions, yet at the current step they have received a miscoordination penalty due to the exploration of an arbitrary agent. Then smaller update steps for the three matching agents

would benefit the coordinated learning process of the whole group, since agreement on this optimal joint action would be much easier to be reached than any other optimal joint actions. Therefore, adaptively coordinating over the agent policies and properly assigning credits among the agents are crucial for the agents to achieve efficient coordination in this kind of game.

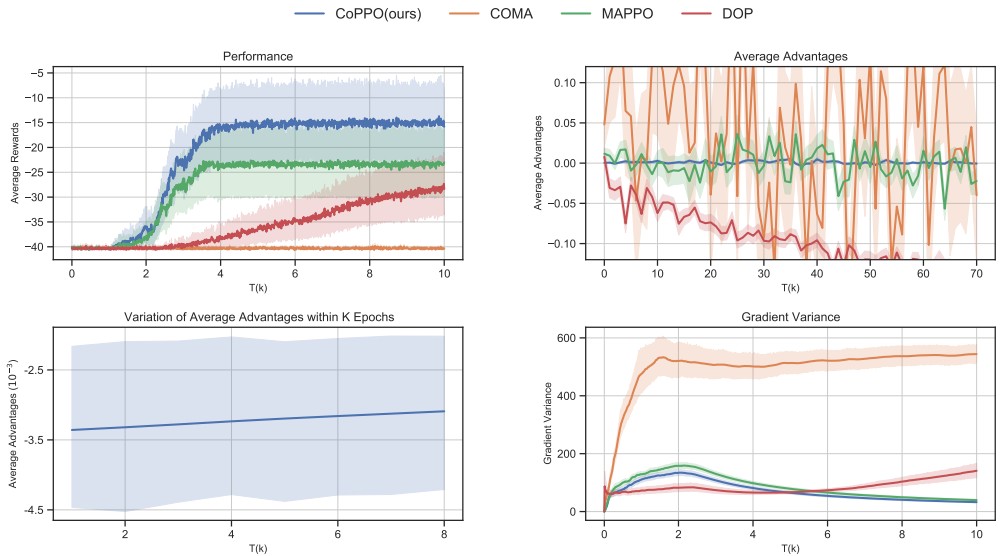

Figure 1: Upper left: average rewards; Upper right: average advantages after a penalty; Lower left: the variation of the average advantages within $K$ (here $K = 8$) optimization epochs every time after a penalty; Lower right: running policy gradient variance.

We train CoPPO, COMA (Foerster et al., 2018), MAPPO (Yu et al., 2021) and DOP (Wang et al., 2020) for 10,000 timesteps, and the final results are averaged over 100 runs. The hyperparameters and other implementation details are described in Appendix C.1. From Fig. 1-upper left, we can see that CoPPO significantly outperforms other methods in terms of average rewards. Fig. 1-upper right presents an explanation of the result by showing the local advantages averaged among the three matching agents every time after receiving a miscoordination penalty. Each point on the horizontal axis represents a time step after a miscoordination penalty. While the times of penalties in a single run vary for different algorithms, we take the minimum times of 70 over all runs. The vertical axis represents $\frac{1}{3} \sum_{j \neq i} A^j$ for COMA, MAPPO and DOP, and represents the mean of $K$ epochs during one update, i.e., $\frac{1}{3} \sum_{j \neq i} \frac{1}{K} \sum_{k=1}^{K} \tilde{A}_k^j$, for CoPPO ($i$ indicates the unmatching agent). Note that CoPPO can obtain the smallest local advantages that are close to 0 compared to other methods, indicating the smallest step sizes for the three agents in the direction of changing the current action. Fig. 1-lower left shows the overall variation of the average advantages within $K$ optimization epochs after receiving a miscoordination penalty. We can see that as the number of epochs increases, the absolute value of the average advantage of the three matching agents gradually decreases by considering the update of other agents. Since the absolute value actually determines the step size of update, a smaller value indicates a small adaptation in their current actions. This is consistent with what we have discussed in Section 3.3. Fig. 1-lower right further implies that through this dynamic process, the agents succeed in learning to coordinate their update steps carefully, yielding the smallest gradient variance among the four methods.

**Ablation study 1**   Fig. 2 provides an ablation study of the double clipping trick. We can see that a proper intermediate inner clipping threshold improves the global performance, and the double clipping trick indeed reduces the variance of the policy gradient. In contrast to DOP, which achieves low gradient variance at the expense of lack of direct coordination over the policies, CoPPO can strike a nice balance between reducing variance and achieving coordination, by taking other agents' policy update into consideration. To make our results more convincing, experiments on more cooperative matrix games with different varieties are also conducted in Appendix D.1.

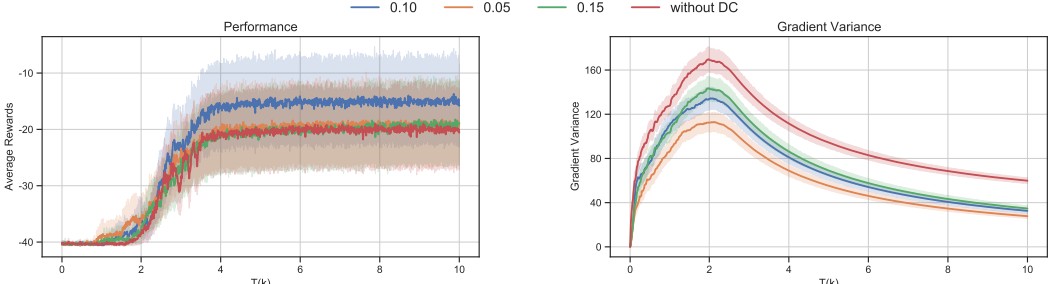

Figure 2: Ablation study of the double clipping trick. The numbers $0.05, 0.10, 0.15$ represent the inner clipping threshold, and "without DC" represents the case when the trick is not used. Left: average rewards; Right: running policy gradient variance.

## 4.2 StarCraft II

We evaluate CoPPO in SMAC against various state-of-the-art methods, including policy-based methods (COMA (Foerster et al., 2018), MAPPO (Yu et al., 2021) and DOP (Wang et al., 2020)) and value-based methods (QMIX (Rashid et al., 2018) and QTRAN (Son et al., 2019)). The implementation of these baselines follows the original versions. The win rates are tested over 32 evaluation episodes after each training iteration. The hyperparameter settings and other implementation details are presented in Appendix C.2. The results are averaged over 6 different random seeds for easy maps (the upper row in Fig. 3), and 8 different random seeds for harder maps (the lower row in Fig. 3). Note that CoPPO outperforms several strong baselines including the latest multi-agent PPO (i.e., MAPPO) method in SMAC across various types and difficulties, especially in **Hard** (3s5z, 10m_vs_11m) and **Super Hard** (MMM2) maps. Moreover, as an on-policy method, CoPPO shows better stability across different runs, which is indicated by a narrower confidence interval around the learning curves.

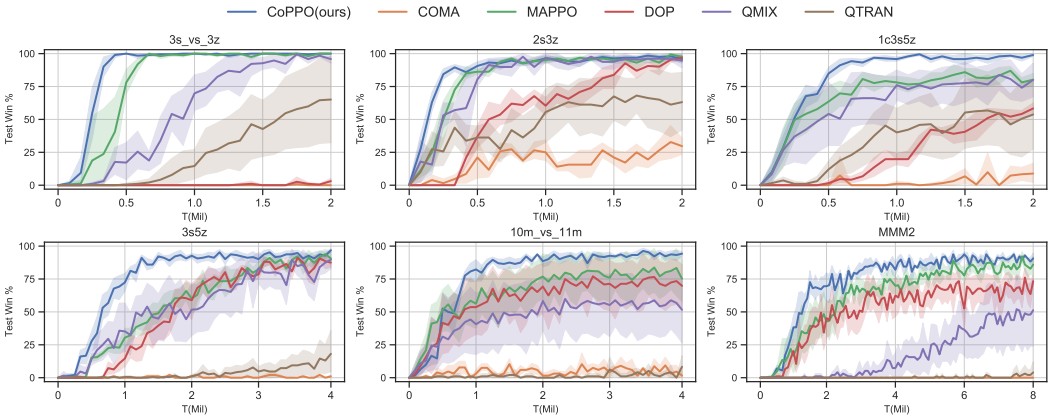

Figure 3: Comparisons against baselines on SMAC.

**Ablation study 2** The first row in Fig. 4 shows the ablation study of double clipping in SMAC, and we can see that the results share the same pattern as in Section 4.1.

**Ablation study 3** In Section 3.2, the global advantage is decomposed into a weighted sum of local advantages. We also compare it to a mixing network with non-negative weights and the results are shown Fig. 4. Similar to QMIX (Rashid et al., 2018), the effectiveness of the mixing network may largely owe to the improvement in the representational ability for the global advantage function.

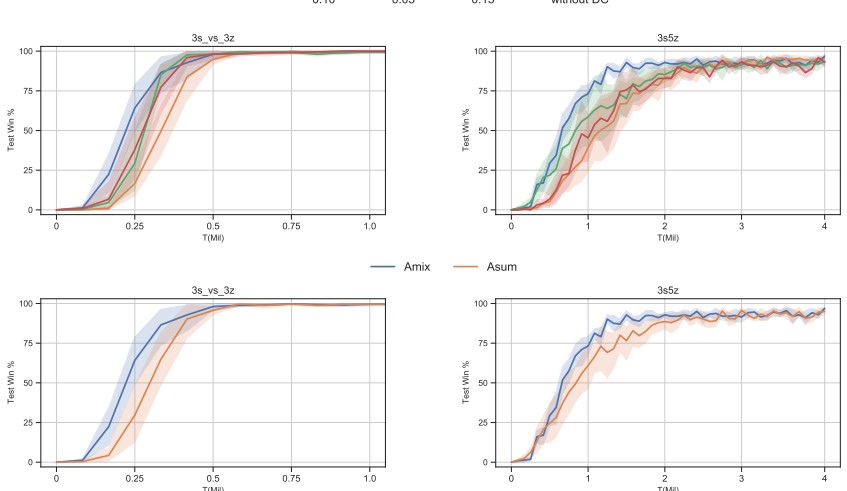

Figure 4: Ablation studies on the double clipping (the upper row) and the way of advantage decomposition (the lower row), evaluated on two maps respectively. In the upper row, the numbers "0.05, 0.10, 0.15" represent the values of the inner clipping threshold, and "without DC" represents the case where the double clipping trick is not utilized. In the lower row, "Amix" refers to a non-negative-weighted neural network, and "Asum" refers to an arithmetic summation.

## 5 Related Work

In recent years, there has been significant progress in CoMARL. Fully centralized methods suffer from scalability issues due to the exponential growth of joint action space, and applying DQN to each agent independently while treating the other agents as part of environment (Tampuu et al., 2017) suffers from the non-stationary issue (Hernandez-Leal et al., 2017; Papoudakis et al., 2019). The CTDE paradigm (Foerster et al., 2016) reaches a compromise between centralized and decentralized approaches, assuming a laboratory setting where each agent's policy can be trained using extra global information while maintaining scalable decentralized execution.

A series of work have been developed in the CTDE setting, including both value-based and policy-based methods. Most of the value-based MARL methods estimate joint action-value function by mixing local value functions. VDN (Sunehag et al., 2018) first introduces value decomposition to make the advantage of centralized training and mixes the local value functions via arithmetic summation. To improve the representational ability of the joint action-value function, QMIX (Rashid et al., 2018) proposes to mix the local action-value functions via a non-negative-weighted neural network. QTRAN (Son et al., 2019) studies the decentralization & suboptimatlity trade-off and introduces a corresponding penalty term in the objective to handle it, which further enlarges the class of representable value functions. As for the policy-based methods, COMA (Foerster et al., 2018) presents the counterfactual advantage to address the credit assignment issue. MADDPG (Lowe et al., 2017) extends DDPG (Lillicrap et al., 2015) by learning centralized value functions which are conditioned on additional global information, such as other agents' actions. DOP (Wang et al., 2020) introduces value decomposition into the multi-agent actor-critic framework, which enables off-policy critic learning and addresses the centralized-decentralized mismatch issue. MAPPO (Yu et al., 2021) generalizes PPO (Schulman et al., 2017) to multi-agent settings using a global value function.

The most relevant works are MAPPO (Yu et al., 2021), MATRPO (Li and He, 2020) and MATRL (Wen et al., 2020). MAPPO extends PPO to multi-agent settings simply by centralizing the critics. With additional techniques such as Value Normalization, MAPPO achieves promising performance compared to several strong baselines. Note that our implementation is built on the one of MAPPO (please refer to Appendix C.2 for more details).

As for MATRPO and MATRL, they both try to extend TRPO (Schulman et al., 2015a) to multi-agent settings. MATRPO focuses on fully decentralized training, which is realized through splitting the joint TRPO objective into $N$ independent parts for each agent and transforming it into a consensus optimization problem; while MATRL computes independent trust regions for each agent assuming

that other agents' policies are fixed, and then solves a meta-game in order to find the best response to the predicted joint policy of other agents derived by independent trust region optimization. Different from our work, they adopt the settings where agents have separate local reward signals. By comparison, CoPPO does not directly optimize the constrained objective derived in Section 3.1, but instead incorporates the trust region constraint into the optimization objective, in order to reduce the computational complexity and simplify the implementation. CoPPO can sufficiently take advantage of the centralized training and enable a coordinated adaptation of step size among agents during the policy update process.

## 6  Conclusion

In this paper, we extend the PPO algorithm to the multi-agent setting and propose an algorithm named CoPPO through a theoretically-grounded derivation that ensures approximately monotonic policy improvement. CoPPO can properly address the issues of scalability and credit assignment, which is interpreted both theoretically and empirically. We also introduce a double clipping trick to strike the balance between reducing variance and achieving coordination by considering other agents' update. Experiments on specially designed cooperative matrix games and the SMAC benchmark verify that CoPPO outperforms several strong baselines and is competitive with the latest multi-agent methods.

## Acknowledgments and Disclosure of Funding

The work is supported by the National Natural Science Foundation of China (No. 62076259 and No. 62076263) and the Tencent AI Lab Rhino-Bird Focused Research Program (No. JR202063). The authors would like to thank Wenxuan Zhu for pointing out some of the mistakes in the proof, Xingzhou Lou and Xianjie Zhang for running some of the experiments, as well as Siling Chen for proofreading the manuscript. Finally, the reviewers and metareviewer are highly appreciated for their constructive feedback on the paper.

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
