## A  Mathematical Details

### A.1  Difference between the performance of two joint policies

In Section 3.1, the difference between the performance of two joint policies is expressed as follows:

$$J(\tilde{\boldsymbol{\pi}}) - J(\boldsymbol{\pi}) = \mathbb{E}_{\boldsymbol{a} \sim \tilde{\boldsymbol{\pi}}, s \sim \rho^{\tilde{\boldsymbol{\pi}}}} \left[ A^{\boldsymbol{\pi}}(s, \boldsymbol{a}) \right], \tag{11}$$

where $\rho^{\tilde{\boldsymbol{\pi}}}$ is the unnormalized discounted visitation frequencies, i.e. $\sum_{t=0}^{\infty} \gamma^t \sum_s \Pr(s_t = s | \tilde{\boldsymbol{\pi}})$. The proof is a multi-agent version of the proof in (Kakade and Langford, 2002). Now we provide the mathematical detail formally.

*Proof.*

$$J(\tilde{\boldsymbol{\pi}}) - J(\boldsymbol{\pi}) = \mathbb{E}_{\tilde{\boldsymbol{\pi}}} \left[ \sum_{t=0}^{\infty} \gamma^t R_{t+1} - V^{\boldsymbol{\pi}}(s_0) \right] \tag{12}$$

$$= \mathbb{E}_{\tilde{\boldsymbol{\pi}}} \left[ R_1 + \gamma V^{\boldsymbol{\pi}}(s_1) - V^{\boldsymbol{\pi}}(s_0) + \gamma [R_2 + \gamma V^{\boldsymbol{\pi}}(s_2) - V^{\boldsymbol{\pi}}(s_1)] + \cdots \right] \tag{13}$$

$$= \mathbb{E}_{\tilde{\boldsymbol{\pi}}} \left[ \sum_{t=0}^{\infty} \gamma^t A^{\boldsymbol{\pi}}(s_t, \boldsymbol{a}) \right] \tag{14}$$

$$= \sum_{t=0}^{\infty} \gamma^t \sum_s \Pr(s_t = s | \tilde{\boldsymbol{\pi}}) \sum_{\boldsymbol{a}} \tilde{\boldsymbol{\pi}}(\boldsymbol{a}|s) \left[ Q^{\boldsymbol{\pi}}(s, \boldsymbol{a}) - V^{\boldsymbol{\pi}}(s) \right] \tag{15}$$

$$= \mathbb{E}_{\boldsymbol{a} \sim \tilde{\boldsymbol{\pi}}, s \sim \rho^{\tilde{\boldsymbol{\pi}}}} \left[ A^{\boldsymbol{\pi}}(s, \boldsymbol{a}) \right] \tag{16}$$

$\square$

### A.2  Approximation that matches the true value to first order

In Section 3.1, we claim that $\tilde{J}_{\boldsymbol{\pi}}(\tilde{\boldsymbol{\pi}})$ matches $J(\tilde{\boldsymbol{\pi}})$ to first order. Intuitively, this means that a sufficiently small update of the joint policy which improves $\tilde{J}_{\boldsymbol{\pi}}(\tilde{\boldsymbol{\pi}})$ will also improve $J(\tilde{\boldsymbol{\pi}})$. Now we prove it formally.

*Proof.* We represent the policy using its parameter, i.e. $\theta$ for $\boldsymbol{\pi}$ and $\tilde{\theta}$ for $\tilde{\boldsymbol{\pi}}$. Because $\tilde{J}_{\boldsymbol{\pi}}(\boldsymbol{\pi}) = J(\boldsymbol{\pi})$, there are $\tilde{J}_{\theta}(\theta) = J(\theta)$. Furthermore, we have:

$$\nabla_{\tilde{\theta}} \tilde{J}_{\theta}(\tilde{\theta})\big|_{\theta} = \nabla_{\tilde{\theta}} \left( J(\theta) + \mathbb{E}_{\boldsymbol{a} \sim \tilde{\boldsymbol{\pi}}, s \sim \rho^{\tilde{\boldsymbol{\pi}}}} \left[ A^{\boldsymbol{\pi}}(s, \boldsymbol{a}) \right] \right) \tag{17}$$

$$= \sum_t \gamma^t \sum_s \Pr(s_t = s | \boldsymbol{\pi}) \sum_{\boldsymbol{a}} \nabla_{\tilde{\theta}} \tilde{\boldsymbol{\pi}}(\boldsymbol{a}|s)\big|_{\theta} A^{\boldsymbol{\pi}}(s, \boldsymbol{a}) \tag{18}$$

$$= \nabla_{\tilde{\theta}} J(\tilde{\boldsymbol{\pi}})\big|_{\theta}, \tag{19}$$

where the last step is indicated by Theorem 1 in (Sutton et al., 2000).

$\square$

### A.3  Upper bound for the error of joint policy approximation

**Theorem.** *Let* $\epsilon = \max_{s, \boldsymbol{a}} |A^{\boldsymbol{\pi}}(s, \boldsymbol{a})|$, $\alpha_i = \sqrt{\frac{1}{2} D_{TV}^{\max}[\pi^i || \tilde{\pi}^i]}$, $1 \leq i \leq N$, *and* $N$ *be the total number of agents, then the error of the approximation in Eq. 4 can be explicitly bounded as follows:*

$$\left| J(\tilde{\boldsymbol{\pi}}) - \tilde{J}_{\boldsymbol{\pi}}(\tilde{\boldsymbol{\pi}}) \right| \leq 4\epsilon \left[ \frac{1 - \gamma \prod_{i=1}^N (1 - \alpha_i)}{1 - \gamma} - 1 \right]. \tag{20}$$

*Proof.* We first prove that for a fixed $s$, the following inequality holds:

$$\left| \mathbb{E}_{\boldsymbol{a} \sim \tilde{\boldsymbol{\pi}}} \left[ A^{\boldsymbol{\pi}}(s, \boldsymbol{a}) \right] \right| \leq 2\epsilon \left[ 1 - \prod_{i=1}^N (1 - \alpha_i) \right]. \tag{21}$$

Note that

$$\mathbb{E}_{\boldsymbol{a}\sim\boldsymbol{\pi}}[A^{\boldsymbol{\pi}}(s,\boldsymbol{a})] = \boldsymbol{\pi}(\boldsymbol{a}|s)\left[Q(s,\boldsymbol{a}) - V(s)\right] \tag{22}$$
$$= V(s) - V(s) \tag{23}$$
$$= 0. \tag{24}$$

Therefore,

$$\mathbb{E}_{\tilde{\boldsymbol{a}}\sim\tilde{\boldsymbol{\pi}}}[A^{\boldsymbol{\pi}}(s,\tilde{\boldsymbol{a}})] = \mathbb{E}_{(\boldsymbol{a},\tilde{\boldsymbol{a}})\sim(\boldsymbol{\pi},\tilde{\boldsymbol{\pi}})}[A^{\boldsymbol{\pi}}(s,\tilde{\boldsymbol{a}}) - A^{\boldsymbol{\pi}}(s,\boldsymbol{a})] \tag{25}$$
$$= \Pr(\boldsymbol{a}\neq\tilde{\boldsymbol{a}})\cdot\mathbb{E}_{(\boldsymbol{a},\tilde{\boldsymbol{a}})\sim(\boldsymbol{\pi},\tilde{\boldsymbol{\pi}})}[A^{\boldsymbol{\pi}}(s,\tilde{\boldsymbol{a}}) - A^{\boldsymbol{\pi}}(s,\boldsymbol{a})] \tag{26}$$
$$= \left[1 - \prod_{i=1}^{N}\left(1 - \Pr\left(a^i\neq\tilde{a}^{-i}\right)\right)\right]\mathbb{E}_{(\boldsymbol{a},\tilde{\boldsymbol{a}})\sim(\boldsymbol{\pi},\tilde{\boldsymbol{\pi}})}[A^{\boldsymbol{\pi}}(s,\tilde{\boldsymbol{a}}) - A^{\boldsymbol{\pi}}(s,\boldsymbol{a})] \tag{27}$$
$$\leq \left[1 - \prod_{i=1}^{N}(1 - \eta_i)\right]\mathbb{E}_{(\boldsymbol{a},\tilde{\boldsymbol{a}})\sim(\boldsymbol{\pi},\tilde{\boldsymbol{\pi}})}[A^{\boldsymbol{\pi}}(s,\tilde{\boldsymbol{a}}) - A^{\boldsymbol{\pi}}(s,\boldsymbol{a})] \tag{28}$$
$$\leq \left[1 - \prod_{i=1}^{N}(1 - \eta_i)\right]\cdot 2\max_{s,\boldsymbol{a}}|A^{\boldsymbol{\pi}}(s,\boldsymbol{a})| \tag{29}$$
$$= 2\epsilon\left[1 - \prod_{i=1}^{N}(1 - \eta_i)\right], \tag{30}$$

where $\eta_i = \max_{\tau^i}\Pr(a^i\neq\tilde{a}^i|\tau^i)$, and $(\boldsymbol{\pi},\tilde{\boldsymbol{\pi}})$ represents $\{(\pi^1,\tilde{\pi}^1),\ldots,(\pi^N,\tilde{\pi}^N)\}$, $(\pi^i,\tilde{\pi}^i)$ is an $\alpha_i$-coupled policy pair for $i = 1,2,\ldots,N$. The definition of $\alpha_i$-coupled policy pair in (Schulman et al., 2015a) implies that $(\pi^i,\tilde{\pi}^i)$ is a joint distribution $p(a^i,\tilde{a}^i|\tau^i)$ satisfying $\Pr(a^i\neq\tilde{a}^i|\tau^i)\leq\alpha_i$.

From Proposition 4.7 in (Levin and Peres, 2017), if we have two distributions $p_X, p_Y$ that satisfy $D_{TV}(p_X\|p_Y) = \alpha$, then there exists a joint distribution $P(X,Y)$ whose marginals are $p_X, p_Y$, such that:

$$\Pr(X = Y) = 1 - \alpha \tag{31}$$

Furthermore, note that there is a relationship between the total variation divergence and the KL divergence (Pollard, 2000): $D_{TV}(p\|q)^2 \leq \frac{1}{2}D_{KL}(p\|q)$. Now let $\alpha_i = \max_{\tau^i}\sqrt{\frac{1}{2}D_{KL}\left[\pi^i(\cdot|\tau^i)\|\tilde{\pi}^i(\cdot|\tau^i)\right]}$, then there exists a joint distribution $(\pi^i,\tilde{\pi}^i)$ whose marginals are $\pi^i,\tilde{\pi}^i$, satisfying:

$$\Pr(a^i = \tilde{a}^i|\tau^i) \geq 1 - \alpha_i. \tag{32}$$

Thus $\eta_i \leq \alpha_i$. Since $\eta_i, \alpha_i \leq 1$, $\left[1 - \prod_{i=1}^{N}(1 - \eta_i)\right]$ will increase as $\eta_i$ increases. Then Eq. (21) can be derived by replacing $\eta_i$ with $\alpha_i$ in Eq. (30).

For simplification, we denote $\mathbb{E}_{\tilde{\boldsymbol{a}}\sim\tilde{\boldsymbol{\pi}}}[A^{\boldsymbol{\pi}}(s,\tilde{\boldsymbol{a}})]$ as $\bar{A}^{\tilde{\boldsymbol{\pi}},\boldsymbol{\pi}}(s)$ and use $n_t$ to represent the times $\boldsymbol{a}\neq\tilde{\boldsymbol{a}}$ before timestep $t$. Then there is:

$$\left|\mathbb{E}_{s_t\sim\rho^{\tilde{\boldsymbol{\pi}}}}[\bar{A}^{\tilde{\boldsymbol{\pi}},\boldsymbol{\pi}}(s_t)] - \mathbb{E}_{s_t\sim\rho^{\boldsymbol{\pi}}}[\bar{A}^{\tilde{\boldsymbol{\pi}},\boldsymbol{\pi}}(s_t)]\right| \tag{33}$$
$$= \Pr(n_t > 0)\cdot\left|\mathbb{E}_{s_t\sim\rho^{\tilde{\boldsymbol{\pi}}}}[\bar{A}^{\tilde{\boldsymbol{\pi}},\boldsymbol{\pi}}(s_t)] - \mathbb{E}_{s_t\sim\rho^{\boldsymbol{\pi}}}[\bar{A}^{\tilde{\boldsymbol{\pi}},\boldsymbol{\pi}}(s_t)]\right| \tag{34}$$
$$= (1 - \Pr(n_t = 0))\cdot\left|\mathbb{E}_{s_t\sim\rho^{\tilde{\boldsymbol{\pi}}}|n_t>0}[\bar{A}^{\tilde{\boldsymbol{\pi}},\boldsymbol{\pi}}(s_t)] - \mathbb{E}_{s_t\sim\rho^{\boldsymbol{\pi}}|n_t>0}[\bar{A}^{\tilde{\boldsymbol{\pi}},\boldsymbol{\pi}}(s_t)]\right| \tag{35}$$
$$= \left(1 - \prod_{t'=0}^{t}\prod_{i=1}^{N}\Pr(a_t^i = \tilde{a}_t^i|\tau^i)\right)\cdot|\cdots| \tag{36}$$
$$\leq \left(1 - \prod_{i=1}^{N}(1 - \alpha_i)^t\right)\cdot|\cdots| \tag{37}$$
$$\leq \left(1 - \prod_{i=1}^{N}(1 - \alpha_i)^t\right)\cdot 2\max_{s}\left|\bar{A}^{\tilde{\boldsymbol{\pi}},\boldsymbol{\pi}}(s)\right|, \tag{38}$$

where $|\cdots|$ denotes $\left|\mathbb{E}_{s_t \sim \rho^{\tilde{\pi}}|n_t>0}[\bar{A}^{\tilde{\pi},\pi}(s_t)] - \mathbb{E}_{s_t \sim \rho^{\pi}|n_t>0}[\bar{A}^{\tilde{\pi},\pi}(s_t)]\right|$ for brevity. Then, the following can be derived using Eq. (21):

$$\left|\mathbb{E}_{s_t \sim \rho^{\tilde{\pi}}}[\bar{A}^{\tilde{\pi},\pi}(s_t)] - \mathbb{E}_{s_t \sim \rho^{\pi}}[\bar{A}^{\tilde{\pi},\pi}(s_t)]\right| \tag{39}$$

$$\leq 2\left(1 - \prod_{i=1}^{N}(1-\alpha_i)^t\right)\left[1 - \prod_{i=1}^{N}(1-\alpha_i)\right]\max_{s,\boldsymbol{a}}|A^{\pi}(s,\boldsymbol{a})| \tag{40}$$

$$\leq 4\epsilon\left[1 - \prod_{i=1}^{N}(1-\alpha_i)\right]\left[1 - \prod_{i=1}^{N}(1-\alpha_i)^t\right], \tag{41}$$

Finally we reach our conclusion:

$$\left|J(\tilde{\pi}) - L_{\pi}(\tilde{\pi})\right| = \left|\mathbb{E}_{\boldsymbol{a}\sim\tilde{\pi},s\sim\rho^{\tilde{\pi}}}[A^{\pi}(s,\boldsymbol{a})] - \mathbb{E}_{\boldsymbol{a}\sim\tilde{\pi},s\sim\rho^{\pi}}[A^{\pi}(s,\boldsymbol{a})]\right| \tag{42}$$

$$= \left|\sum_s \sum_{t=0}^{\infty} \gamma^t \Pr(s_t = s|\tilde{\pi})\sum_{\boldsymbol{a}}\tilde{\pi}(\boldsymbol{a}|s)A^{\pi}(s,\boldsymbol{a}) - \right.$$

$$\left.\sum_s \sum_{t=0}^{\infty} \gamma^t \Pr(s_t = s|\pi)\sum_{\boldsymbol{a}}\tilde{\pi}(\boldsymbol{a}|s)A^{\pi}(s,\boldsymbol{a})\right| \tag{43}$$

$$\leq \sum_{t=0}^{\infty} \gamma^t \left|\mathbb{E}_{s_t \sim \rho^{\tilde{\pi}}}[\bar{A}^{\tilde{\pi},\pi}(s_t)] - \mathbb{E}_{s_t \sim \rho^{\pi}}[\bar{A}^{\tilde{\pi},\pi}(s_t)]\right| \tag{44}$$

$$\leq \sum_{t=0}^{\infty} \gamma^t \cdot 4\epsilon\left[1 - \prod_{i=1}^{N}(1-\alpha_i)\right]\left[1 - \prod_{i=1}^{N}(1-\alpha_i)^t\right] \tag{45}$$

$$= 4\epsilon\left[1 - \prod_{i=1}^{N}(1-\alpha_i)\right]\left[\frac{1}{1-\gamma} - \frac{1}{1-\gamma\prod_{i=1}^{N}(1-\alpha_i)}\right] \tag{46}$$

$$\leq 4\epsilon\left[\frac{1 - \gamma\prod_{i=1}^{N}(1-\alpha_i)}{1-\gamma} - 1\right]. \tag{47}$$

$$\square$$

### A.4 Transformation from the joint objective into the local objectives

In Section 3.2, the joint objective is derived as:

$$\operatorname*{maximize}_{\theta^1,\ldots,\theta^N}\ \mathbb{E}_{\boldsymbol{a}\sim\boldsymbol{\pi}_{old}}\left\{\min\left[\left(\prod_{j=1}^{N}r^j\right)A^{\boldsymbol{\pi}}, \operatorname{clip}\left(\left(\prod_{j=1}^{N}r^j\right), 1-\epsilon, 1+\epsilon\right)A^{\boldsymbol{\pi}}\right]\right\}, \tag{48}$$

where $\theta^j$ is the parameter of agent $j$'s policy, and $r^j = \frac{\pi^j(a^j|\tau^j;\theta^j)}{\pi^j_{old}(a^j|\tau^j;\theta^j_{old})}$. After a linear decomposition on $A^{\boldsymbol{\pi}}$ with non-negative weights (i.e. $A^{\boldsymbol{\pi}} = \sum_j c^j A^j$), the objective above then can be transformed into:

$$\operatorname*{maximize}_{\theta^1,\ldots,\theta^N}\ \mathbb{E}_{\boldsymbol{a}\sim\boldsymbol{\pi}_{old}}\left\{\min\left[\left(\prod_{j\neq i}r^j\right)r^i A^i, \operatorname{clip}\left(\left(\prod_{j\neq i}r^j\right)r^i, 1-\epsilon, 1+\epsilon\right)A^i\right]\right\}, \tag{49}$$

where $i = 1,\ldots,N$. Now we provide a detailed proof.

*Proof.* If

$$\min\left[\left(\prod_{j=1}^{N}r^j\right)A^{\boldsymbol{\pi}}, \operatorname{clip}\left(\left(\prod_{j=1}^{N}r^j\right), 1-\epsilon, 1+\epsilon\right)A^{\boldsymbol{\pi}}\right] = \operatorname{clip}\left(\left(\prod_{j=1}^{N}r^j\right), 1-\epsilon, 1+\epsilon\right)A^{\boldsymbol{\pi}},$$
$$\tag{50}$$

then the objective is actually $(1 - \epsilon)A^{\boldsymbol{\pi}}$ or $(1 + \epsilon)A^{\boldsymbol{\pi}}$, and no gradient will be backpropagated as none of $\theta^1, \ldots, \theta^N$ is in the objective. Furthermore, there is

$$\min \left[ \left( \prod_{j=1}^N r^j \right) A^{\boldsymbol{\pi}}, \text{clip} \left( \left( \prod_{j=1}^N r^j \right), 1 - \epsilon, 1 + \epsilon \right) A^{\boldsymbol{\pi}} \right] \tag{51}$$

$$= \min \left[ \sum_i c^i \left( \prod_{j=1}^N r^j \right) A^i, \sum_i c^i \text{clip} \left( \left( \prod_{j=1}^N r^j \right), 1 - \epsilon, 1 + \epsilon \right) A^i \right]. \tag{52}$$

Thus, the discussion can be simplified to the case where

$$\min \left[ \left( \prod_{j=1}^N r^j \right) A^{\boldsymbol{\pi}}, \text{clip} \left( \left( \prod_{j=1}^N r^j \right), 1 - \epsilon, 1 + \epsilon \right) A^{\boldsymbol{\pi}} \right] = \left( \prod_{j=1}^N r^j \right) A^{\boldsymbol{\pi}}. \tag{53}$$

While $\frac{\partial \left( \prod_{j=1}^N r^j \right) A^{\boldsymbol{\pi}}}{\partial \left( \prod_{j=1}^N r^j \right) A^i} = c^i$ and $c^i \geq 0$, there is

$$\max_{\theta^1, \ldots, \theta^N} \left( \prod_{j=1}^N r^j \right) A^{\boldsymbol{\pi}} = \max_{\theta^1, \ldots, \theta^N} \sum_i c^i \left( \prod_{j=1}^N r^j \right) A^i \tag{54}$$

$$= \sum_i c^i \max_{\theta^1, \ldots, \theta^N} \left( \prod_{j=1}^N r^j \right) A^i \tag{55}$$

Therefore, the transformation from Eq. (48) to Eq. (49) is proved.

$\square$

### A.5 The potential high variance of probability ratio product

Section 3.2 mentions that there exists a risk of high variance in estimating the policy gradient when optimizing Eq. (7), due to the following proposition:

**Proposition.** *Assuming that the agents are fully independent during execution, then the following inequality holds:*

$$Var_{\boldsymbol{a}^{-i} \sim \boldsymbol{\pi}_{old}^{-i}} \left[ \prod_{j \neq i} r^j \right] \geq \prod_{j \neq i} Var_{a^j \sim \pi_{old}^j} \left[ r^j \right], \tag{56}$$

*where $r^j = \frac{\pi^j(a^j | \tau^j; \theta^j)}{\pi_{old}^j(a^j | \tau^j; \theta_{old}^j)}$.*

Because the agents execute the actions based only on locally observable information, it is reasonable to assume that $\pi^i$ and $\pi^j$ is independent when $i \neq j$. Now we present a detailed proof for this proposition.

*Proof.* Because the agents are fully independent during execution, there is a decomposition that $\boldsymbol{\pi}^{-i}(\boldsymbol{a}^{-i} | \boldsymbol{\tau}^{-i}) = \prod_{j \neq i} \pi^j(a^j | \tau^j)$.

Now we use mathematical induction to prove the fact. First, we assume that there are 3 agents, and let $i = 3$ without loss in generality. Then there is:

$$\text{Var}_{a^1, a^2} [r_1 r_2] = \mathbb{E}_{a^1, a^2} \left[ (r_1 r_2)^2 \right] - \left( \mathbb{E}_{a^1, a^2} [r_1 r_2] \right)^2 \tag{57}$$

$$= \mathbb{E}_{a^1} \left[ r_1^2 \right] \mathbb{E}_{a^2} \left[ r_2^2 \right] - \left( \mathbb{E}_{a^1} [r_1] \mathbb{E}_{a^2} [r_2] \right)^2. \tag{58}$$

Hence, there is:

$$\text{Var}_{a^1,a^2}\left[r_1 r_2\right] - \text{Var}_{a^1}\left[r_1\right]\text{Var}_{a^2}\left[r_2\right] \tag{59}$$

$$=\mathbb{E}_{a^1}\left[r_1^2\right]\mathbb{E}_{a^2}\left[r_2^2\right] - \left(\mathbb{E}_{a^1}\left[r_1\right]\mathbb{E}_{a^2}\left[r_2\right]\right)^2 -$$
$$\left[\mathbb{E}_{a^1}\left[r_1^2\right] - \left(\mathbb{E}_{a^1}\left[r_1\right]\right)^2\right]\left[\mathbb{E}_{a^2}\left[r_2^2\right] - \left(\mathbb{E}_{a^2}\left[r_2\right]\right)^2\right] \tag{60}$$

$$= \left(\mathbb{E}_{a^1}\left[r_1\right]\right)^2\mathbb{E}_{a^2}\left[r_2^2\right] + \left(\mathbb{E}_{a^2}\left[r_2\right]\right)^2\mathbb{E}_{a^1}\left[r_1^2\right] - 2\left(\mathbb{E}_{a^1}\left[r_1\right]\mathbb{E}_{a^2}\left[r_2\right]\right)^2 \tag{61}$$

$$= \left(\mathbb{E}_{a^1}\left[r_1\right]\right)^2\text{Var}_{a^2}\left[r_2\right] + \left(\mathbb{E}_{a^2}\left[r_2\right]\right)^2\text{Var}_{a^1}\left[r_1\right] \geq 0. \tag{62}$$

By now we have proven $\text{Var}_{a^1,a^2}\left[r_1 r_2\right] \geq \text{Var}_{a^1}\left[r_1\right]\text{Var}_{a^2}\left[r_2\right]$. Then if Eq. (56) holds for the case of $N$ agents, then obviously there is:

$$\prod_{j\neq i}^{N+1}\text{Var}\left[r^j\right] = \left(\prod_{j\neq i}^{N}\text{Var}\left[r^j\right]\right)\text{Var}_{a^{N+1}}\left[r^{N+1}\right] \tag{63}$$

$$\leq \text{Var}\left[\prod_{j\neq i}^{N} r^j\right]\text{Var}_{a^{N+1}}\left[r^{N+1}\right] \tag{64}$$

$$\leq \text{Var}\left[\prod_{j\neq i}^{N+1} r^j\right], \tag{65}$$

thus proving the proposition.

$\square$

## A.6 The simplification in the analysis of CoPPO and MAPPO

In Section 3.3, the difference between CoPPO and MAPPO is simplified to the difference between $\mathbb{E}_{\boldsymbol{\pi}_{old}}\left[r_k^i A^i\right]$ and $\mathbb{E}_{\boldsymbol{\pi}_{old}}\left[r_k^i \tilde{A}_k^i\right]$. Now we detail the rationality of this simplification.

In each update, the value of both the two objectives start from the respective lower bounds and are updated conservatively during the optimization epochs. The objectives monotonically increase or decrease until they reach the clipping threshold. No update will be made when the objective is clipped, because $\theta^i$ is not in the clipped value (i.e. $(1-\epsilon_1)A^i$ or $(1+\epsilon_1)A^i$) and no gradient will be backpropagated then, just as discussed in Appendix A.4.

## A.7 $\prod_{j\neq i} r_k^j$ implies the variation of the probability to take $a^{-i}$

Section 3.3 mentions that $\prod_{j\neq i} r_k^j > 1$ will cause an increase in $\boldsymbol{\pi}^{-i}(\boldsymbol{a}^{-i}|\boldsymbol{\tau}^{-i})$ and vice versa. Now we provide the details.

Similar to Appendix A.5, the decentralized policies can be viewed independently, thus $\boldsymbol{\pi}^{-i}(\boldsymbol{a}^{-i}|\boldsymbol{\tau}^{-i}) = \prod_{j\neq i}\pi^j(a^j|\tau^j)$. By definition, $\prod_{j\neq i} r_k^j = \prod_{j\neq i}\frac{\pi_k^j(a^j|\tau^j)}{\pi_{old}^j(a^j|\tau^j)}$. Synthesizing the two equations, we have $\prod_{j\neq i} r_k^j = \frac{\boldsymbol{\pi}_k^{-i}(\boldsymbol{a}^{-i}|\boldsymbol{\tau}^{-i})}{\boldsymbol{\pi}_{old}^{-i}(\boldsymbol{a}^{-i}|\boldsymbol{\tau}^{-i})}$ which suggests that if $\prod_{j\neq i} r_k^j > 1$, $\boldsymbol{a}^{-i}$ will be more likely to be jointly performed by the other agents given similar observations, and vice versa.

## B Pseudo Code

The details of our CoPPO algorithm are given in Algorithm 1.

## C Implementation Details

Experiments are conducted on NVIDIA Quadro RTX 5000 GPUs. The network architectures, optimizers, hyperparameters and environment settings in the cooperative matrix game and SMAC are described respectively in the following subsections.

**Algorithm 1** The CoPPO Algorithm
___
1: Initialize policies $\pi_{old}^1, \ldots, \pi_{old}^N$ for $N$ agents respectively;
2: **for** $iteration = 1, 2, \ldots$ **do**
3:     **for** $rollout\ thread = 1, 2, \ldots, R$ **do**
4:         Run policies $\pi_{old}^{1:N}$ in environment for $T$ time steps;
5:         Compute advantage estimates $\hat{A}_{1:T}^{\pi_{old}^j}, \ldots, \hat{A}_{1:T}^{\pi_{old}^j}$, $j = 1, 2, \ldots, N$;
6:     **end for**
7:     **for** $k = 0, 1, \ldots, K - 1$ **do**
8:         **for** $i = 1, 2, \ldots, N$ **do**
9:             Optimize the objective
10:             $L(\theta^i) = \mathbb{E}_{\boldsymbol{a} \sim \boldsymbol{\pi}_{old}} \left\{ \min \left[ g(\boldsymbol{r}^{-i}) r^i A^i, \mathrm{clip}\left( g(\boldsymbol{r}^{-i}) r^i, 1 - \epsilon_1, 1 + \epsilon_1 \right) A^i \right] \right\}$
11:             to update the policy $\pi^i$ w.r.t. $\theta^i$;
12:         **end for**
13:     **end for**
14:     $\theta_{old}^j \leftarrow \theta_K^j$, $j = 1, 2, \ldots N$;
15: **end for**
___

## C.1   Cooperative matrix game

We utilize the same actor-critic network architecture for all the algorithms. The actor consists of two 18-dimensional fully-connected layers with $\tanh$ activation. For the critic, two 72-dimensional fully-connected layers are adopted with $\tanh$ activation. For the hyper network in DOP which is used to derive the weights and biases for local value mixing, we use two 36-dimensional fully-connected layers with $\tanh$ activation for both the weights and biases deriving. The optimization of both the actors and critics is conducted using RMSprop with the learning rate of $5 \times 10^{-4}$ and $\alpha$ of 0.99. No momentum or weight decay is used in the optimizers. The discounted factor is set to 0.99; the number of the optimization epochs (i.e. $K$) for CoPPO and MAPPO is set to 8; the outer clipping threshold (i.e. $\epsilon$ for MAPPO and $\epsilon_1$ for CoPPO) is set to 0.20. For the inner clipping threshold in CoPPO, we consider $\epsilon_2 \in \{0.05, 0.10, 0.15\}$ and adopt 0.10 in the comparison with baselines. For exploration, we use $\epsilon$-greedy with $\epsilon$ annealed linearly from 0.9 to 0.02 over $6k$ timesteps.

## C.2   SMAC

The same actor-critic network architecture are utilized for all maps we have evaluated on. Both the actor and critic networks consist of two fully-connected layers, one GRU layer and one fully-connected layer sequentially with ReLU activation. For the mixing network mentioned in Section 3.2, we adopt the hyper network in (Rashid et al., 2018) to derive the weights and bias for local advantages, and enforce the weights to be non-negative. Similar to QMIX, the input of the hyper network is the global state. The dimensions of these layers are all set to 64, except for the 32-dimensional hidden layers of the mixing network.

For the evaluation on different maps, all the hyperparameters are fixed except for the number of optimization epochs which is set to 15 for 2s3z, 3s_vs_3z, and 1c3s5z, 10 for 3s5z and 10m_vs_11m, and 8 for MMM2. The number of epochs overall decreases as the difficulty of the map increases, ranging from 5 to 15. The optimization of both the actors and critics is conducted using Adam with the learning rate of $5 \times 10^{-4}$ and optimizer epsilon of $1 \times 10^{-5}$. No weight decay is used in the optimizers. The discounted factor $\gamma$ is set to 0.99. For advantage estimation, the generalized advantage estimation (Schulman et al., 2015b) is adopted and the corresponding hyperparameter $\lambda$ is set to 0.90. Note that state value functions instead of state-action value functions are estimated in SMAC. The inner clipping threshold (i.e. $\epsilon_2$ for CoPPO) is set to 0.10, while the outer clipping threshold (i.e. $\epsilon$ for MAPPO and $\epsilon_1$ for CoPPO) is set to 0.20. 8 parallel environments are run for data collecting.

Overall, our implementation builds upon the one of (Yu et al., 2021). Note that MAPPO uses hand-coded states (i.e. Feature-Pruned Agent-Specific Global State) as the input of value functions, while in our implementation these states are modified into the concatenation of the Environment-Provided Global State and the Local Observation, in order to make the comparison with baselines fair. For the

other baselines, we adopt the official implementations and their default hyperparameter settings that have been fine-tuned on this benchmark.

# D Additional Results

## D.1 Cooperative matrix games

Section 4.1 shows the results on a modification of the two-player penalty game. Now we present the results on other matrix games across different types and different difficulties in Fig. 5, and CoPPO outperforms the other methods in almost all the games, thus showing the general effectiveness. For evaluation, the results are also averaged over 100 runs.

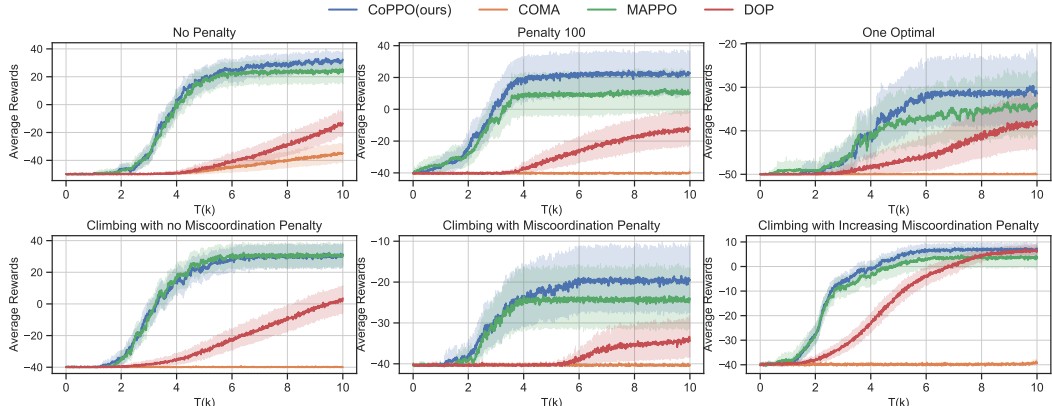

Figure 5: Performance comparisons in six matrix games.

These games are all 4-agent, 9-action cooperative games. The respective reward settings are as follows. The "miscoordination" mentioned below all refers to the case where any three agents act the same while the other does not. Fig. 5-upper left and middle are both simplifications of the penalty game presented in Section 4.1. In Fig. 5-upper left, there is no penalty for miscoordination; in Fig. 5-upper middle, the team reward becomes larger (100) when the agents play the same action. The other rewards are set the same with the one in Section 4.1. In Fig. 5-upper right, there is only one optimal joint action and the difficulty lies mainly in exploration. The agents will receive the reward of 50 if agent $i$ plays action $i$ and -50 otherwise. Fig. 5-lower left is the result on a modification of the *climbing game* that has been used as another challenging test bed for CoMARL algorithms (Claus and Boutilier, 1998), where the reward is $i \cdot 10$ if the agents all play action $i$ and -40 otherwise. Fig. 5-lower middle and right gradually increase the difficulty of the climbing game by setting obstacles in the way of climbing. In Fig. 5-lower middle, the agents will be punished by -50 for miscoordination. As for Fig. 5-lower right, the miscoordination penalty increases as the matching reward increases, i.e. $-i \cdot 10$ for miscoordination on action $i$, hence the risk will become higher and higher when the agents are "climbing" to the optimal joint action.

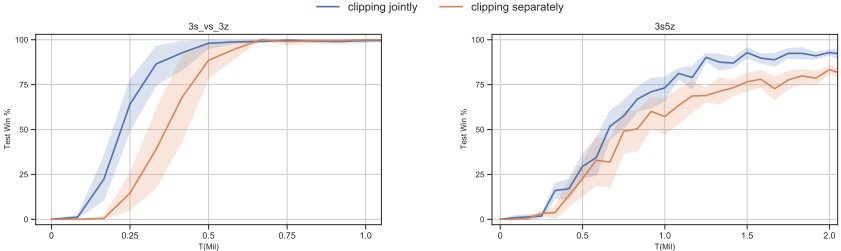

Figure 6: Ablation study on the methods of clipping.

## D.2 SMAC

### D.2.1 Comparison of clipping jointly and separately

We empirically evaluate two clipping approaches mentioned in Section 3.2, i.e. clipping jointly $(\text{clip}(\prod_{j=1}^{N} r^j, \cdot, \cdot))$ and clipping separately $(\prod_{j=1}^{N} \text{clip}(r^j, \cdot, \cdot))$. The results shown in Fig. 6 demonstrate that clipping separately performs worse than clipping jointly. To find the cause resulting in this performance discrepancy, an empirical analysis is conducted on the value of policy gradients and ratio products w.r.t. the two clipping methods, and the results are presented in Fig. 7. Obviously clipping jointly yields more stable ratio product and policy gradients than clipping separately, implying that the performance discrepancy might be owing to the stability in the policy update.

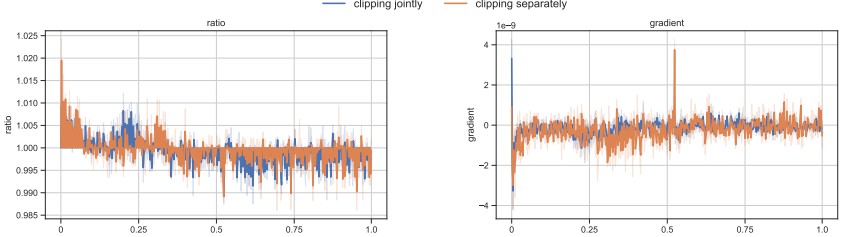

Figure 7: Comparison of two clipping methods on ratio product and mean policy gradients, evaluated on 3s_vs_3z.

### D.2.2 Results on three more maps of SMAC

Some additional results for further verification of the effectiveness of CoPPO in SMAC are given in Fig. 8. Note that CoPPO outperforms all the baselines in the maps we have evaluated on, except for the MMM map where CoPPO achieves competitive performance against MAPPO.

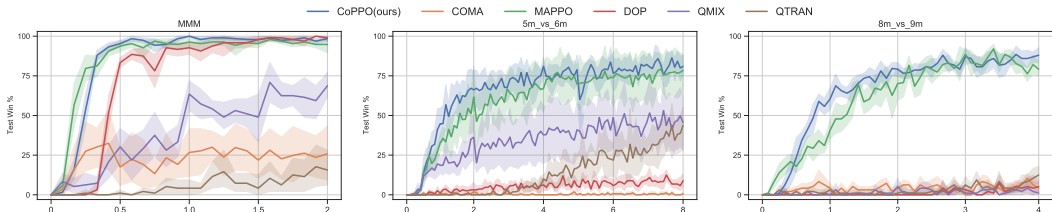

Figure 8: Additional results on SMAC.