# OpenReview forum: "Coordinated Proximal Policy Optimization"
_NeurIPS.cc/2021/Conference — NeurIPS 2021 Poster_

### Official Review · Reviewer_A4sq · 2021-07-03

**Rating:** 7
**Confidence:** 3

**Summary:**

This work presents an extension of the popular proximal policy optimization (PPO) algorithm for cooperative multi-agent tasks.

**Limitations And Societal Impact:**

On the subset of tasks from SMAC selected for experimentation, the proposed methods performs well. I imagine that on at least some tasks from SMAC, the proposed method does not outperform the baselines, so it would be good to know what these are and what the authors' hypotheses are for the lack of improvement in those settings , if they exist.

**Main Review:**

# Strengths
* The proposed method falls into a promising direction for cooperative MARL methods. Much of the focus in this field has been in value-based methods, but simple extensions of single-agent policy-optimization methods have recently been shown to be competitive with the SOTA value-based ones. As such, this work is timely.
* The exposition is clear and succinct. Theoretical contributions are well motivated and explained intuitively.
* The expected baselines are present in the experiments, and the proposed method reliably outperforms them.

# Weaknesses
I don't believe the paper has any serious shortcomings, though there are a few minor points that could be addressed.
* The colors should be consistent in Figures 1 and 2. In other words, whichever setting for the inner clipping threshold that is used in Figure 1 should be blue in figure 2. Perhaps the other settings should use colors or line patterns that aren't used in figure 1.
* It would be good to see how CoPPO performs in more of the Hard and (especially) Super Hard tasks from SMAC. The MAPPO paper performs experiments on the whole set of tasks.

# Overall Thoughts
This paper is a well thought out and sensible follow up to recent works which show the relatively untapped potential of policy-based methods for cooperative MARL.

**Time Spent Reviewing:**

2

---

> ### Author Response · Authors · 2021-08-10
> **Author Response to Reviewer A4sq**
>
> We thank the reviewer for the valuable comments.  Please see the response below.
>
> **Q1: "The colors should be consistent in Figures 1 and 2. "**
>
> **A1:** Thanks for your suggestion. We have re-plotted Figure 2 and will replace it in the future version.
>
> **Q2: "It would be good to see how CoPPO performs in more of the Hard and (especially) Super Hard tasks from SMAC. "**
>
> **A2:** According to the comments of **reviewer ubaz**, we have additionally evaluated CoPPO and all the baselines in MMM (Easy), 10m\_vs\_11m (Hard) and MMM2 (Super Hard) to further verify the effectiveness of CoPPO. The results in the following table show the test win rate averaged over 6 different random seeds at corresponding timesteps (the left most column). These results overall verify the robustness of CoPPO across various map types and difficulties in SMAC, except in the easy map MMM where CoPPO learns slightly slower than MAPPO.
>
> - **MMM**
>
>   | T(Mil) |  CoPPO   |  MAPPO   | QMIX | QTRAN |   DOP    | COMA |
>   | :----: | :------: | :------: | :--: | :---: | :------: | :--: |
>   |  0.25  |   0.51   | **0.75** | 0.02 | 0.00  |   0.45   | 0.38 |
>   |  0.50  | **0.94** | **0.94** | 0.09 | 0.00  |   0.91   | 0.52 |
>   |  0.75  | **0.96** |   0.95   | 0.27 | 0.06  | **0.96** | 0.61 |
>   |  1.00  | **0.99** |   0.96   | 0.56 | 0.13  | **0.99** | 0.65 |
>
> - **10m\_vs\_11m**
>
>   | T(Mil) |  CoPPO   | MAPPO | QMIX | QTRAN | DOP  | COMA |
>   | :----: | :------: | :---: | :--: | :---: | :--: | :--: |
>   |  0.5   | **0.52** | 0.51  | 0.05 | 0.01  | 0.07 | 0.06 |
>   |  1.0   | **0.87** | 0.54  | 0.07 | 0.03  | 0.55 | 0.06 |
>   |  1.5   | **0.92** | 0.66  | 0.12 | 0.03  | 0.60 | 0.08 |
>   |  2.0   | **0.92** | 0.73  | 0.18 | 0.08  | 0.66 | 0.00 |
>   |  2.5   | **0.96** | 0.78  | 0.33 | 0.06  | 0.69 | 0.03 |
>   |  3.0   | **0.96** | 0.78  | 0.45 | 0.03  | 0.74 | 0.08 |
>   |  3.5   | **0.94** | 0.83  | 0.51 | 0.01  | 0.79 | 0.00 |
>   |  4.0   | **0.96** | 0.75  | 0.67 | 0.07  | 0.78 | 0.06 |
>
> - **MMM2**
>
>   | T(Mil) |  CoPPO   | MAPPO | QMIX | QTRAN |   DOP    | COMA |
>   | :----: | :------: | :---: | :--: | :---: | :------: | :--: |
>   |   1    |   0.14   | 0.10  | 0.00 | 0.00  | **0.20** | 0.00 |
>   |   2    | **0.55** | 0.37  | 0.00 | 0.00  |   0.48   | 0.00 |
>   |   3    | **0.73** | 0.60  | 0.06 | 0.00  |   0.59   | 0.00 |
>   |   4    | **0.83** | 0.70  | 0.13 | 0.00  |   0.63   | 0.00 |
>   |   5    | **0.86** | 0.77  | 0.27 | 0.00  |   0.74   | 0.00 |
>   |   6    | **0.90** | 0.82  | 0.46 | 0.00  |   0.79   | 0.00 |
>   |   7    | **0.90** | 0.84  | 0.66 | 0.02  |   0.83   | 0.00 |
>   |   8    | **0.91** | 0.86  | 0.71 | 0.02  |   0.81   | 0.00 |
>
> We believe that these maps are representative enough to support the superior performance of CoPPO. If the reviewer feel that more Hard and Super Hard maps are needed to make the final decision, we will evaluate CoPPO in more maps and add the results into the final version of our paper.

---

### Official Review · Reviewer_iDqu · 2021-07-16

**Rating:** 5
**Confidence:** 4

**Summary:**

This paper proposes a new CTDE algorithm CoPPO for cooperative MARL. CoPPO is based on the extended objective of TRPO in MARL settings. CoPPO uses the clip objective of PPO to implement the trust region constraint of TRPO and proposes a double clipping trick to adjust the step sizes of different agents.

**Limitations And Societal Impact:**

As stated above

**Main Review:**

The double clip trick is interesting, and it also has the efficacy of dynamic credit assignment. However, I have the following concerns.

1. The objective of CoPPO is actually a direct extension of TRPO in multi-agent settings. If we take a centralized view, suppose the policies of agents are mutually independent, following the conclusion of TRPO, we can direct derive the same result. So the main contribution of this paper may lie in the double clipping trick for optimizing the objective.

2. The paper is unclear in many ways  at least from the writing. For example, is a joint Q(s,a)  learned to compute A^i? where does c^i come from?

3. Morever, the empirical results on SMAC maps seem to have some problems. The performance of MAPPO in the Hard and Super Hard maps are much lower than the results in the Table 1 in the original paper of MAPPO. And the learning curves of MAPPO in these maps are different with the original paper.

4. Moreover, the performance of CoPPO in these maps is lower than the results of MAPPO in the original paper.

I expect these concerns can be addressed during rebuttal.

**Time Spent Reviewing:**

3 hours

---

> ### Author Response · Authors · 2021-08-10
> **Author Response to Reviewer iDqu**
>
> We thank the reviewer for all the valuable comments. Please see the response below.
>
> **Q1: "The objective of CoPPO is actually a direct extension of TRPO in multi-agent settings."**
>
> **A1:** We agree that the derivation process of Theorem 1 and the objective in Eq. (7) follow the existing TRPO and PPO papers. However, in order to cope with the issues arising in multi-agent settings, which is the major concern of our work, we provided further transformations over Eq. (7) in order to derive our final objective in Eq. (10) and gave theoretical analysis of why these transformations work: *1)* we decomposed the global objective into local ones for better credit assignment and formally proved its soundness in Appendix A.4;  *2)* we proved that when local advantage $A^i$​ is negative, the variance incurred by the other agents grows at least exponentially w.r.t. the number of agents (i.e., Appendix A.5), and proposed a double clipping trick to handle this issue. Moreover,  we provided intuitive interpretation of our final objective being able to realize a dynamic credit assignment through balancing the variance and the coordination between the agents (i.e., Section 3.3), and verified this interpretation empirically on modified matrix  games (i.e., Section 4.1).  We consider the combination of all the above aspects to constitute the final contribution of our paper.
>
> **Q2: "the main contribution of this paper may lie in the double clipping trick for optimizing the objective."**
>
> **A2:** Besides the clarification in **A1**, we should also note that the highlight of our work lies in the dynamic coordination in the step size adjustment of the agents' policies. The double clipping technique here copes with the variance issue incurred by the other agents, and thus playing an auxiliary role in improving the final performance of CoPPO, which can be also observed in the ablation study (i.e., Section 4.2) in the paper.
>
>
>
> **Q3: "Is a joint Q(s,a) learned to compute A^i? Where does c^i come from?"**
>
> **A3:** Thanks for pointing out the confusing notations in our paper. Regarding the decomposition of the joint advantage, we mean that each agent learns a respective value function to compute $A^i$​. The non-negative $c^i$​ can be learned using a hyper network like Figure 1 in the DOP paper [1]. It is worth noting that here $c^i$​​​​​ is used just for expression brevity since our final algorithm instead takes advantage of a mixing network to aggregate the local advantage values.
>
> **Q4: "The performance of MAPPO in the Hard and Super Hard maps are much lower than the results in the Table 1 in the original paper of MAPPO. And the learning curves of MAPPO in these maps are different with the original paper.."**
>
> **A4:** Considering that some methods can converge to a similar level provided enough training steps, in our paper, we focus on the learning curves of different methods in order to compare their convergence speed and thus learning efficiency. However,  we failed to reproduce the learning curves of MAPPO in three maps (i.e., 3s5z, 5m\_vs\_6m, 3s5z\_vs\_3s6z) after great efforts, even following exactly the same parameter settings of the original MAPPO paper. We guess this might be due to the high randomness of these maps or some other implementation issues. We will contact the authors of MAPPO after the review process in order to further check the reasons. Nevertheless, by comparing the learning curves in the original MAPPO paper and our results, we can still observe that CoPPO converges faster than MAPPO in these maps  (please see **A5** for more details). Please note that, besides the above implementation issue, another cause of the performance gap in Table 1 in the MAPPO paper is due to the fewer training steps in our experiments. Anyway, we will fix this consistency issue after the reviewing process and will make necessary modifications  in our future version.
>
> **Q5: "The performance of CoPPO in these maps is lower than the results of MAPPO in the original paper."**
>
> **A5:** We check the test win rate in the original MAPPO paper (Figure 13) every 1M timesteps (0.5M for 3s5z) and present a point-to-point comparison of these two methods in the following table for a better illustration. From the results we can see that CoPPO generally outperforms, or at least performs similarly with, MAPPO in these maps. Please note that the values of MAPPO are roughly estimated from the original figures, showing the overall convergence trend of MAPPO. As stated ahead, we will fix the consistency issue of performance of MAPPO later and provide more accurate comparision between CoPPO and MAPPO in the final version of our paper.
>
> - **3s5z**
>
>   | T(Mil) |  CoPPO   |  MAPPO   |
>   | :----: | :------: | :------: |
>   |  0.5   |   0.31   | **0.35** |
>   |  1.0   | **0.74** |   0.4    |
>   |  1.5   | **0.89** |   0.81   |
>   |  2.0   | **0.92** |   0.85   |
>   |  2.5   | **0.93** |   0.85   |
>   |  3.0   |   0.90   | **0.93** |
>   |  3.5   | **0.95** |   0.91   |
>   |  4.0   |   0.97   | **1.0**  |
>
> - **5m\_vs\_6m**
>
>   | T(Mil) |  CoPPO   | MAPPO |
>   | :----: | :------: | :---: |
>   |   1    | **0.38** | 0.17  |
>   |   2    | **0.56** | 0.29  |
>   |   3    | **0.66** | 0.45  |
>   |   4    | **0.74** | 0.51  |
>   |   5    | **0.74** | 0.60  |
>   |   6    | **0.75** | 0.61  |
>   |   7    | **0.77** | 0.66  |
>   |   8    | **0.76** | 0.65  |
>
> - **3s5z\_vs\_3s6z**
>
>   | T(Mil) |  CoPPO   |  MAPPO   |
>   | :----: | :------: | :------: |
>   |   1    | **0.13** |   0.08   |
>   |   2    | **0.38** |   0.22   |
>   |   3    | **0.62** |   0.61   |
>   |   4    | **0.70** | **0.70** |
>   |   5    | **0.75** | **0.75** |
>   |   6    | **0.79** |   0.78   |
>   |   7    | **0.81** |   0.79   |
>   |   8    | **0.84** |   0.81   |
>
> ------
>
> ref.
>
> [1] Wang et al., Off-policy multi-agent decomposed policy gradients, ICLR 2021.

---

> > ### Comment · Reviewer_iDqu · 2021-08-27
> > **Further comments**
> >
> > 1. The performance of MAPPO in 5m_vs_6m is different from the results in the paper (Figure 4, https://arxiv.org/pdf/2103.01955.pdf). The win rate of MAPPO at 8Mil is about 85%, not 65% as you reported.
> > 2. I also looked at the results you presented in the responses to other reviewers. It seems not fair for baselines. For example, in 10m_vs_11m, QMIX is much higher than 67% (as reported in MAPPO); in MMM2, MAPPO can reach 90% with 8Mil, but your result is only 86%.
> > 3. Moreover, why is not QPLEX chosen as a baseline?

---

> > > ### Author Response · Authors · 2021-08-28
> > > **Further Response**
> > >
> > > 1. We carefully check the paper from the link you have provided and find that the authors of MAPPO have redone the experiments and updated the result of 5m\_vs\_6m. The new arxiv version was uploaded on July 5th, which is after the submission deadline of NeurlPS. However, when preparing our paper, we referred to the previous arxiv version of MAPPO (https://arxiv.org/pdf/2103.01955v1.pdf), where the performance of MAPPO in 5m\_vs\_6m is around 65%.
> > >
> > >    Due to the complexity and randomness of SMAC, achieving SOTA performance in all maps is extremely difficult. Methods such as QMIX, QTRAN, MAVEN and QPLEX also fail to guarantee completely dominant performance. Nevertheless, despite of 5m\_vs\_6m where CoPPO slightly underperforms the updated version of MAPPO, CoPPO still outperforms or at least performs similarly with the baselines in the remaining maps.
> > >
> > >    In order to keep consistency with the above result, we will modify our previous claim that "CoPPO achieves state-of-the-art performance against several strong baselines in the StarCraft II micromanagement tasks." to "**CoPPO outperforms several strong baselines and is competitive with the latest multiagent PPO (i.e., MAPPO) method in SMAC across various types and difficulties.**" in the future version.
> > >
> > > 2. As for the inconsistency of the results of some baselines, we should clarify that our result of QMIX in 10m\_vs\_11m (67%) is consistent with the result reported in some previous papers such as DOP. It seems that the higher performance of QMIX (90%) reported in MAPPO is due to the input modified global state to the mixer network (i.e., QMix (MG) therein). Even in this case, our final performance (96%) is still higher than the result reported in MAPPO.  As for the minor performance gap between 86% and 90%, we think it reasonable considering the inherent randomness especially in Super Hard maps. In all, we will spare no efforts to keep consistency in the results of all the baseline methods, and make necessary update in the future version of our paper.
> > >
> > > 3. The latest version of MAPPO shows that MAPPO generally outperforms QPLEX. Considering that our method can also outperform MAPPO in most maps, we can expect that our method would achieve higher performance than QPLEX. In order to verify this claim, following your advice, we will add these results and provide rigorous comparison  in our future version.

---

### Official Review · Reviewer_yRTL · 2021-07-16

**Rating:** 5
**Confidence:** 4

**Summary:**

This paper presents a multi-agent generalization of proximal policy optimization algorithm (PPO) (Schulman'17). The authors provide monotonic policy improvement guarantee and use double clipping technique (Ye'20) in order to mitigate the exploration issues caused by multiplication of importance weights in off-policy sampling. They also interpret this objective function as dynamic credit assignment among agents which can reduce the variance in update. They test the proposed approach on cooperative matrix game and Starcraft tasks to show its outperformance over other multi agent RL algorithms.

**Main Review:**

The flow of the paper is nice and readable. The idea of the paper on multi agent RL with PPO is interesting as well.
In summary, I see this paper as combination PPO and double clipping technique(Ye'20) in multiagents. Because the derivations are pretty straightforward and similar to  TRPO and PPO papers. Overall, The results are nice but not surprising at all.
Some questions:
Does double clipping affect the convergence or policy improvement lemma? Can clipping be applied separately to each importance weight to avoid the problem?

**Time Spent Reviewing:**

3

---

> ### Author Response · Authors · 2021-08-10
> **Author Response to Reviewer yRTL**
>
> We thank the reviewer for all these valuable comments. We provide point-by-point responses below.
>
> **Q1: "In summary, I see this paper as combination PPO and double clipping technique(Ye'20) in multiagents".**
>
> **A1:** First, we would like to point out that, although building upon PPO,  CoPPO involves several novel modifications and theoretical analysis that are specifically tailored to multi-agent settings (please refer to **A2** for more explanations).
>
> In terms of the dual-clipping trick in [1], we should clarify that the goal and implementation of our double clipping technique are both different from theirs. First,  our double clipping targets at reducing the variance incurred by the ratio product of other agents, and thus striking a balance between the variance and coordination in multi-agent settings. In contrast, dual clipping is applied to handle the policy deviation arising from adopting PPO in a distributed, off-policy setting. Second, in terms of implementation, our double clipping employs an inner clipping on the ratio products of other agents, while dual clipping sets a lower bound to the entire objective instead of the probability ratio and thus performing an additional "outer clipping" in each individual PPO objective of the distributed roll-out actors. Please also note that the highlight of our work lies in the dynamic coordination in the step size adjustment of the agents' policies, and thus the double clipping technique here plays an auxiliary role in improving the final performance of CoPPO, which can be also observed in the ablation study in the paper.
>
>
>
> **Q2: "the derivations are pretty straightforward and similar to TRPO and PPO papers".**
>
> **A2:** We agree that the derivation process of Theorem 1 and the objective in Eq. (7) follow the existing TRPO and PPO papers. However, in order to cope with the issues arising in multi-agent settings, which is the major concern of our work, we provided several transformations over Eq. (7) in order to derive our final objective in Eq. (10) and gave theoretical analysis of why these transformations work: **1)** we decomposed the global objective into local ones for better credit assignment and formally proved its soundness in Appendix A.4;  **2)** we proved that when local advantage $A^i$​​​​ is negative, the variance incurred by the other agents grows at least exponentially w.r.t. the number of agents (i.e., Appendix A.5), and proposed a double clipping trick to handle this issue. **Moreover**,  we provided intuitive interpretation of our final objective being able to realize a dynamic credit assignment through balancing the variance and the coordination between the agents (i.e., Section 3.3), and verified this interpretation empirically on modified matrix games (i.e., Section 4.1). We consider the combination of all the above aspects to constitute the final contribution of our paper.
>
> **Q3: "Does double clipping affect the convergence or policy improvement lemma?"**
>
> **A3:** The result in Theorem 1 overall indicates optimizing over $ J_{\boldsymbol{\pi}} (\tilde{\boldsymbol{\pi}})$ while limiting the update of policies, and Eq. (7) combines these two goals into a single objective using a clipping technique to control $\prod_{j=1}^N r^j$. As a modification of Eq. (7), double clipping can be seen as dividing the original clipping into two phases: limiting $\prod_{j\neq i} r^j$ to $[1-\epsilon_2, 1+\epsilon_2]$ and later limiting $r^i$ to $[\frac{1-\epsilon_1}{1+\epsilon_2}, \frac{1+\epsilon_1}{1-\epsilon_2}]$. Since this acts the same as not using double clipping from the perspective of limiting $\prod_{j=1}^N r^j$​​, and only changes the priority of limiting these ratios, double clipping would not affect the policy improvement lemma in essence.
>
> **Q4: "Can clipping be applied separately to each importance weight to avoid the problem?"**
>
> **A4:** According to your comment, we empirically tested the idea of applying clipping separately to each importance ratio. We respectively chose 0.05, 0.10, 0.20 and 0.25 as the clipping threshold, and the results show that clipping separately performed much worse than clipping jointly. From the theoretical perspective, the two methods both aim to restrict the update of policies and reduce the variance incurred by other agents. However, intuitively, it is easier to confine the output value when clipping jointly, i.e., $\text{clip}(\prod_{j=1}^Nr^j,\cdot,\cdot)$​​​​​, than clipping seperately, i.e., $\prod_{j=1}^N\text{clip}(r^j,\cdot,\cdot)$​​​​​. To verify this claim, we conducted an empirical analysis on the value of policy gradients and the ratio products after using these two different clipping methods. We found that clipping separately yielded much larger ratio products and gradients than clipping jointly, implying that the performance discrepancy might be owing to the unstability in the policy update. We will add more analysis about this issue in the final version of our paper.
>
> ------
>
> ref.
>
> [1] Ye et al., Mastering complex control in moba games with deep reinforcement learning, AAAI2020.

---

### Official Review · Reviewer_ubaz · 2021-07-16

**Rating:** 6
**Confidence:** 5

**Summary:**

This paper proposes the coordinated proximal policy optimization (CoPPO) algorithm for cooperative multi-agent systems. CoPPO is a novel algorithm with centralized training and decentralized execution (CTDE) which achieves better coordination between agents via a coordinated adapting process of the agents' step size when updating policies. The empirical results in penalty games and SMAC show that CoPPO outperforms the SOTA algorithms such as MAPPO and DOP.

**Limitations And Societal Impact:**

See the above

**Main Review:**

The paper is technically sound and well written. I think it is a good paper. However, I still have one concern which I think is quite important for making the final decision: Why did the author only run 6 maps in the SMAC experiments? Why not compare the algorithms in more SMAC maps such as MMM, MMM2, 10m_vs_11m?

**Time Spent Reviewing:**

6

---

> ### Author Response · Authors · 2021-08-10
> **Author Response to Reviewer ubaz**
>
> We thank the reviewer for the valuable comments.
>
> Originally, we considered the maps in our evaluation to be representative enough to cover both types of scenario and levels of difficulty: (1) *heterogeneous & symmetric* (2s3z, 1c3s5z, 3s5z), *homogeneous & asymmetric* (5m\_vs\_6m), *heterogeneous & asymmetric* (3s5z\_vs\_3s6z) and *micro-trick: kiting* (3s\_vs\_3z); (2) Easy (2s3z, 1c3s5z, 3s\_vs\_3z), Hard (3s5z, 5m\_vs\_6m) and Super Hard (3s5z\_vs\_3s6z). Nevertheless, according to your suggestion, we have additionally evaluated CoPPO and all the baselines in MMM (Easy), 10m\_vs\_11m (Hard) and MMM2 (Super Hard) to further verify the effectiveness of CoPPO (due to the presentation issue in the rebuttal system, we only provide numerical results, but all the new results will be added into our paper later in the form of learning curves). The following tables show the results of median test win rate at corresponding timesteps (the left most column), which are averaged over 6 different random seeds. These results overall verify the robustness of CoPPO across various map types and difficulties in SMAC, except in the easy map MMM where CoPPO learns a bit slower than MAPPO.
>
> - **MMM**
>
>   | T(Mil) |  CoPPO   |  MAPPO   | QMIX | QTRAN |   DOP    | COMA |
>   | :----: | :------: | :------: | :--: | :---: | :------: | :--: |
>   |  0.25  |   0.51   | **0.75** | 0.02 | 0.00  |   0.45   | 0.38 |
>   |  0.50  | **0.94** | **0.94** | 0.09 | 0.00  |   0.91   | 0.52 |
>   |  0.75  | **0.96** |   0.95   | 0.27 | 0.06  | **0.96** | 0.61 |
>   |  1.00  | **0.99** |   0.96   | 0.56 | 0.13  | **0.99** | 0.65 |
>
> - **10m\_vs\_11m**
>
>   | T(Mil) |  CoPPO   | MAPPO | QMIX | QTRAN | DOP  | COMA |
>   | :----: | :------: | :---: | :--: | :---: | :--: | :--: |
>   |  0.5   | **0.52** | 0.51  | 0.05 | 0.01  | 0.07 | 0.06 |
>   |  1.0   | **0.87** | 0.54  | 0.07 | 0.03  | 0.55 | 0.06 |
>   |  1.5   | **0.92** | 0.66  | 0.12 | 0.03  | 0.60 | 0.08 |
>   |  2.0   | **0.92** | 0.73  | 0.18 | 0.08  | 0.66 | 0.00 |
>   |  2.5   | **0.96** | 0.78  | 0.33 | 0.06  | 0.69 | 0.03 |
>   |  3.0   | **0.96** | 0.78  | 0.45 | 0.03  | 0.74 | 0.08 |
>   |  3.5   | **0.94** | 0.83  | 0.51 | 0.01  | 0.79 | 0.00 |
>   |  4.0   | **0.96** | 0.75  | 0.67 | 0.07  | 0.78 | 0.06 |
>
> - **MMM2**
>
>   | T(Mil) |  CoPPO   | MAPPO | QMIX | QTRAN |   DOP    | COMA |
>   | :----: | :------: | :---: | :--: | :---: | :------: | :--: |
>   |   1    |   0.14   | 0.10  | 0.00 | 0.00  | **0.20** | 0.00 |
>   |   2    | **0.55** | 0.37  | 0.00 | 0.00  |   0.48   | 0.00 |
>   |   3    | **0.73** | 0.60  | 0.06 | 0.00  |   0.59   | 0.00 |
>   |   4    | **0.83** | 0.70  | 0.13 | 0.00  |   0.63   | 0.00 |
>   |   5    | **0.86** | 0.77  | 0.27 | 0.00  |   0.74   | 0.00 |
>   |   6    | **0.90** | 0.82  | 0.46 | 0.00  |   0.79   | 0.00 |
>   |   7    | **0.90** | 0.84  | 0.66 | 0.02  |   0.83   | 0.00 |
>   |   8    | **0.91** | 0.86  | 0.71 | 0.02  |   0.81   | 0.00 |

---

### Decision · Program_Chairs · 2021-09-27

**Decision:**

Accept (Poster)

**Comment:**

The reviewers and AC discussed the paper. There was agreement that the approach is interesting and the results are impressive. The author response was also helpful in addressing some of the concerns of the reviewers (e.g., additional experimental results). There are still some concerns about the novelty of the approach and the comparisons with MAPPO. These should be clarified.